# Detecting Hidden Confounding In Observational Data Using Multiple Environments

**Rickard K.A. Karlsson**
Department of Intelligent Systems
Delft University of Technology
The Netherlands
r.k.a.karlsson@tudelft.nl

**Jesse H. Krijthe**
Department of Intelligent Systems
Delft University of Technology
The Netherlands
j.h.krijthe@tudelft.nl

## Abstract

A common assumption in causal inference from observational data is that there is no hidden confounding. Yet it is, in general, impossible to verify this assumption from a single dataset. Under the assumption of independent causal mechanisms underlying the data-generating process, we demonstrate a way to detect unobserved confounders when having multiple observational datasets coming from different environments. We present a theory for testable conditional independencies that are only absent when there is hidden confounding and examine cases where we violate its assumptions: degenerate & dependent mechanisms, and faithfulness violations. Additionally, we propose a procedure to test these independencies and study its empirical finite-sample behavior using simulation studies and semi-synthetic data based on a real-world dataset. In most cases, the proposed procedure correctly predicts the presence of hidden confounding, particularly when the confounding bias is large.

## 1 Introduction

Estimating the causal effect of a treatment on an outcome is a fundamental challenge in many areas of science and society. While this is straightforwardly done using data from randomized studies, using observational data for this task is appealing since they are often more feasible to collect while also being more representative of the population of interest [Pearl, 2009]. To identify causal effects using such data it is often assumed there is no hidden confounding. When this *untestable* assumption is violated we run the risk of confusing causal relationships with spurious correlations. This can have serious consequences such as unknowingly suggesting a non-effective or, even worse, potentially harmful treatment. Therefore, detecting the presence of hidden confounding is an important problem.

Data collected from different sources often tend to be heterogeneous due to e.g. changing circumstances or time shifts. In this work, we show how such heterogeneity can be exploited to make hidden confounding testable *solely* from observational data. We consider a setting where observational data has been collected from different environments $E$. In each environment, we observe the same treatment $T$ and outcome $Y$, as well as covariates $X$ that are known confounders between $T$ and $Y$. Further, we assume that the data is heterogeneous across these environments under the principle of *Independent Causal Mechanisms* [Peters et al., 2017, Schölkopf et al., 2021], which states that a causal system consists of autonomous modules that do not inform or influence each other. The question we ask is whether there exists further hidden confounding between $T$ and $Y$ after having adjusted for $X$. If that is the case, the causal effect of $T$ on $Y$ is not identifiable in general. Perhaps surprisingly, we demonstrate a way to decide if the causal effect would be identifiable by deriving

37th Conference on Neural Information Processing Systems (NeurIPS 2023).

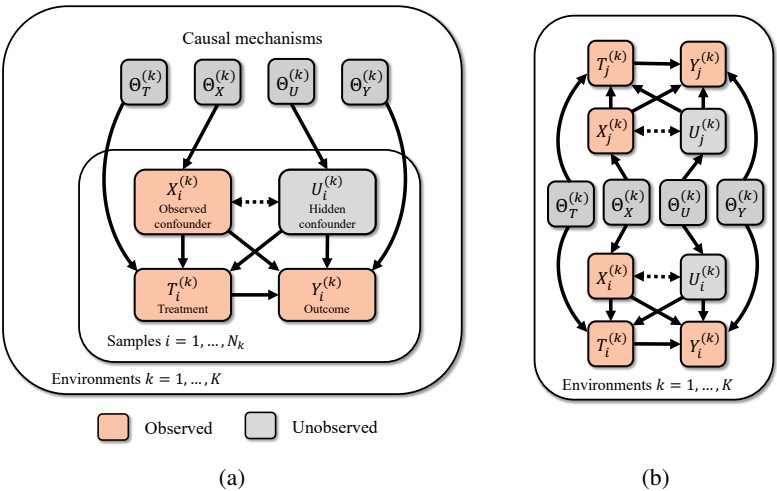

Figure 1: We have multiple observations $i = 1, \ldots, N_k$ of treatment $T_i^{(k)}$, outcome $Y_i^{(k)}$ and confounder $X_i^{(k)}$ in different environments $E^{(k)}$. The dashed bi-directed edge between $X_i^{(k)}$ and $U_i^{(k)}$ allows for any causal relationship, or lack thereof, between the observed and hidden confounder. **(a)**: The hierarchical structure of the multi-environment data; the causal mechanisms are unobserved but we know the indicator $E^{(k)}$ for what environment observations belong to. **(b)**: By unrolling the graph in (a) we see that dependencies exist between any pairs of observations $(i, j)$ from the same environment (when not conditioning on the mechanisms). This can be exploited to detect the presence of the hidden confounder.

testable implications for whether hidden confounding is present or not. We achieve this by exploiting the hierarchical structure of the problem, shown in Figure 1.

As an illustration of a setting where this might be applied, there are many existing multi-level studies in which individuals are nested in clusters and non-randomly assigned to a treatment/control on an individual level. For instance, we can have pre-defined clusters in a multi-level observational study that investigate a specific treatment and outcome from multiple hospitals [Goldstein et al., 2002] or schools [Leite et al., 2015] that care for patients/pupils from different demographics. Here the clusters constitute different environments. Often we might suspect the existence of potential individual-level confounders such as socio-economic status. Now, if these confounding factors have different distributions at each cluster our work proposes a way to statistically test the presence of confounding between the treatment and outcome – even when we do not observe the confounding factors directly.

Another example where our method is suitable is when there are multiple observational studies where no randomized control trials are available, a problem area where systematic procedures are still lacking [Mueller et al., 2018]. In particular, individual participant data meta-analyses are a type of analysis that uses all individual-level data from multiple studies instead of aggregating summary statistics [Riley et al., 2010, Di Angelantonio et al., 2016]. With this information and given that we observe the same treatment and outcome across all studies, our proposed algorithm can also be used to detect if there is common hidden confounding among the studies.

**Contributions** We prove that there exists, under the principle of independent causal mechanisms, testable independencies that are only violated in the presence of unobserved confounding between treatment and outcome (Sec. 4, Theorem 1). Further, we explore the effect of changes and violations of our assumptions – while most assumptions are necessary, we find that some can be relaxed (Sec. 4.1). We then introduce a statistical testing procedure that uses any suitable conditional independence test to detect the presence of hidden confounding in observational datasets from multiple environments (Sec. 4.2). Lastly, we perform an empirical finite-sample analysis of it using both synthetic and semi-synthetic data generated with real-world covariates from the Twins

dataset [Almond et al., 2005, Louizos et al., 2017]. We observe that our proposed procedure correctly predicts the presence of hidden confounding in most cases, particularly when the confounding bias is large (Sec. 5).

## 2 Problem setting

We start with some preliminaries of the causal terminology used in this paper.

**Definition 1** (Causal Graphical Model (CGM)). *A causal graphical model $M = (\mathcal{G}, P)$ over $d$ random variables $\mathbf{V} = (V_1, V_2, \dots, V_d)$ comprises (i) a directed acyclic graph (DAG) $\mathcal{G}$ with vertices $\mathbf{V}$ and edges $V_i \to V_j$ iff $V_i$ is a direct cause of $V_j$, and (ii) a joint distribution $P$ such that it has the following Markov or causal factorization over $\mathcal{G}$:*

$$P(V_1, V_2, \dots, V_d) = \prod_{i=1}^{d} P(V_i \mid \mathrm{Pa}(V_i)) \tag{1}$$

*where $\mathrm{Pa}(V_i)$ denotes the parents (direct causes) of $V_i$ in $\mathcal{G}$ and $P(V_i \mid \mathrm{Pa}(V_i))$ is the causal mechanism of $V_i$.*

The DAG $\mathcal{G}$ encodes various conditional independencies between the variables – also known as d-separations in the DAG, see Pearl [1988, Chapter 3.3] – which we write as $\mathbf{A} \perp\!\!\!\perp_d \mathbf{B} \mid \mathbf{C}$ over some disjoint sets of variables $\mathbf{A}, \mathbf{B}$ and $\mathbf{C}$. We shall assume that conditional independencies in $\mathcal{G}$ imply the same conditional independencies in $P$, and vice versa:

**Assumption 1** (Faithfulness & Causal Markov Property). *For $P$ and $\mathcal{G}$ we have (i) the faithfulness property that $\mathbf{A} \perp\!\!\!\perp_P \mathbf{B} \mid \mathbf{C} \Rightarrow \mathbf{A} \perp\!\!\!\perp_d \mathbf{B} \mid \mathbf{C}$, and (ii) the causal Markov property that $\mathbf{A} \perp\!\!\!\perp_P \mathbf{B} \mid \mathbf{C} \Leftarrow \mathbf{A} \perp\!\!\!\perp_d \mathbf{B} \mid \mathbf{C}$.*

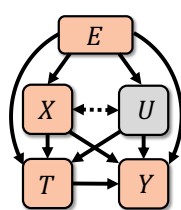

We consider a setting with the following variables in our causal graphical model: a one-dimensional treatment $T \in \mathcal{T}$ and outcome $Y \in \mathcal{Y}$, in addition to some observed covariates $X \in \mathcal{X}$ and unobserved covariates $U \in \mathcal{U}$. We do not restrict the dimensionality of $X$ and $U$. Additionally, in this setting, the environment $E$ has a direct effect on all other variables, making it a root node in $\mathcal{G}$. We say that a variable is a confounder between $T$ and $Y$ if it is a cause of both $T$ and $Y$ in $\mathcal{G}$. We assume that $X$ is a known confounder between $T$ and $Y$, while the relationship between $U$ and the other variables is unknown. Hence, $U$ could be an unobserved hidden confounder (as illustrated in Figure 2) or, for instance, completely unrelated to the other variables.

Figure 2: The setting where we want to detect the presence of a hidden confounder $U$ in $\mathcal{G}$.

Expressed in the framework of Pearl [2009], the goal of causal inference is to estimate the probability $P(Y \mid do(T = t))$ where $do(T = t)$ represents an intervention on the treatment. Without any further assumptions, $P(Y \mid do(T = t))$ is not identifiable from an observational dataset; that is data where we have observed the choice of treatment without influencing it [Pearl, 2009]. In particular, in the setting we consider here, the interventional effect remains unidentifiable if the unobserved $U$ is a confounder between $T$ and $Y$. [1]. Unfortunately, there is no way to check whether such unobserved confounders are present in a single dataset. We will show, however, that things are different when we have access to observational datasets from multiple environments. In this setting, we present a way to detect confounding even if it is not observed, hence demonstrating a novel and valuable approach for verifying an essential prerequisite for causal inference from observational data. In the rest of this section, we present the main assumptions that enable us to do this.

First, we have data from multiple environments $E \in \mathcal{E}$ with different joint distributions $P(T, Y, X, U \mid E)$. We shall use $P_E(\cdot)$ to denote $P(\cdot \mid E)$, and use small letters for the random variables whenever they take particular values. In our setting, we have datasets $D_k = \{t_i^{(k)}, y_i^{(k)}, x_i^{(k)}\}_{i=1}^{N_k}$

---

[1]There exist other procedures that could circumvent this issue, but these alternatives seldom avoid the unconfoundedness assumption completely. As an example, instrumental variable estimation is applicable when there is unobserved confounding between $T$ and $Y$, but only when the relationship between the instrumental variable and $T$ is unconfounded [Angrist et al., 1996].

from multiple environments $e^{(1)}, e^{(2)}, \ldots, e^{(K)}$; each has $N_k$ observations which are assumed to be i.i.d. within the environment. $N_k$ is fixed but can be different for each environment. The environments are related to each other through the following assumption.

**Assumption 2** (Shared Causal Graph). *All environments share the same underlying causal DAG $\mathcal{G}$.*

Next, we specify how changes in $P_E(T, Y, X, U)$ arise between the different environments. We shall assume that the conditional probabilities in (1) – which we refer to as causal mechanisms – vary independently per environment. This is known as the independent causal mechanism principle.

**Assumption 3** (Independent Causal Mechanism (ICM) Principle [Peters et al., 2017]). *The causal generative process of a system's variables is composed of autonomous modules that do not inform or influence each other. In the probabilistic case, this means that the conditional distribution of each variable given its causes (i.e., its parents in the causal graph) does not inform or influence the other mechanisms.*

The above assumption covers two aspects: one concerning *informing* and the other about *influencing*. That the mechanisms do not *inform* each other can be interpreted as that knowing the conditional probability of one variable does not tell us anything about the conditional probabilities of other variables [Janzing and Schölkopf, 2010, Guo et al., 2022]. Further, we assume that changing (or performing an intervention upon) one mechanism has no *influence* on other mechanisms [Schölkopf et al., 2012]. While this notion of independence between mechanisms can be described through a non-stochastic, algorithmic mutual information [Janzing and Schölkopf, 2010], we focus in this work explicitly on statistically independent mechanisms.

To model changes between environments with independent causal mechanisms, we parameterize each causal mechanism with $\Theta_V \in \mathcal{O}_V$ for $V \in \{T, Y, X, U\}$. In each environment, these parameters are fixed and determine the distribution

$$P_E(T, Y, X, U) = \prod_{V \in \{T, Y, X, U\}} P_{\Theta_V}(V \mid \mathrm{Pa}(V)). \tag{2}$$

Note that while we need to know which observations come from which environment, we do not assume to know the particular values of the individual parameters $(\Theta_T, \Theta_Y, \Theta_X, \Theta_U)$ in any environment. We shall assume that environments are randomly sampled from a *distribution over mechanisms* by defining non-degenerate probability measures for each causal mechanism.

**Assumption 4** (Non-degenerate Probabilistic Independent Causal Mechanisms). *The independent causal mechanisms are non-degenerate random variables with probability measures $P(\Theta_V)$ for all $V \in \{T, Y, X, U\}$ such that $\Theta_T$, $\Theta_Y$, $\Theta_X$ and $\Theta_U$ are pairwise independent random variables.*

With the above assumption, when we now say *independent* causal mechanisms, we refer to statistical independence between them. We argue that the above assumption is not particularly strong if we already have Assumption 3; the mechanisms are now allowed to change across environments in a probabilistic manner. Guo et al. [2022] proved the existence of such probability measures for the causal mechanisms when the data comprises an infinitely exchangeable sequence of random variables, drawing parallels to de Finetti's theorem [de Finetti, 1937].

As a final note on the assumptions we have made: these assumptions should not be taken for granted and it is crucial to also understand how violations of them will influence our theory. For this reason, we will cover this topic in Section 4.1.

**Hierarchical model of the environments**   Using these assumptions, we can now express the distribution of the datasets $\{D_k\}_{k=1}^{K}$ as a hierarchical model [Gelman et al., 2013, Chapter 5], wherein we first sample the mechanisms i.i.d. $\Theta_V^{(k)} \sim P(\Theta_V)$ for $k = 1, \ldots, K$ and $V \in \{T, Y, X, U\}$ and then, for each environment $k$, obtain $(T_i^{(k)}, Y_i^{(k)}, X_i^{(k)}, U_i^{(k)})$ by repeatedly sampling $N_k$ times according to (2). Using plate notation, we can compactly represent this hierarchical model in the augmented DAG $\mathcal{G}^*$ shown in Figure 1a. The edges in $\mathcal{G}^*$ between $(T_i^{(k)}, Y_i^{(k)}, X_i^{(k)}, U_i^{(k)})$ are the same as those between $(T, Y, X, U)$ in $\mathcal{G}$ and $\Theta_V^{(k)} \in \mathrm{Pa}(V_i^{(k)})$ where $V \in \{T, Y, X, U\}$, for all $i$ and $k$.

In the next parts of the paper, we will prove how the structure of $\mathcal{G}^*$ implies novel observable constraints in the multi-environment data distribution that can be exploited to statistically test the presence of hidden confounding between $T$ and $Y$ after having adjusted for $X$. But first, we discuss the main literature related to our work.

## 3  Related work

This paper contributes to the growing body of research based on the principle of *Independent Causal Mechanisms* [Peters et al., 2017] which has inspired further research on integrating machine learning and causality [Schölkopf et al., 2012, Peters et al., 2016, von Kügelgen et al., 2020, Schölkopf et al., 2021]. Multiple works have demonstrated how the independent causal mechanism principle could improve causal structure learning when data comes from heterogeneous environments that share the same causal model [Zhang et al., 2017, Ghassami et al., 2018, Guo et al., 2022]. In particular, Guo et al. [2022] demonstrated how independent causal mechanisms imply independence constraints similar to ours when the data is exchangeable – but they assume there exist no unobserved latent variables in contrast to our work where we detect the presence of such variables.

Detecting hidden confounding is hard, and often we can only reason about the plausibility of having unmeasured confounders using some sort of sensitivity analysis [Rosenbaum and Rubin, 1983b, VanderWeele and Ding, 2017, Cinelli et al., 2019]. Other approaches check whether a treatment effect estimate is robust to changes in our assumptions by varying the adjustment set [Lu and White, 2014, Oster, 2019, Su and Henckel, 2022]. However, the guarantees are elusive for whether this type of robustness implies unconfoundedness. Similarly, one could test for heterogeneity of the treatment effect estimates from multiple environments and conclude that if they are different, then it is due to unobserved confounding; this idea bears resemblance to the pseudo-treatment approach discussed by Imbens and Rubin [2015] for assessing unconfoundedness. But testing heterogeneity to detect confounding only works if the treatment effect is assumed to be fixed across all environments, which excludes many real-world settings. Lastly, Janzing and Schölkopf [2018] proposed a method to detect hidden confounding which is restricted to settings with linear models.

In the setting with data from multiple environments, various approaches have been proposed to deal with hidden confounding, typically by combining both experimental and observational data [Bareinboim and Pearl, 2016, Kallus et al., 2018, Athey et al., 2020, Hatt et al., 2022, Ilse et al., 2022, Imbens et al., 2022]. In contrast, we consider a setting combining *only* observational data from multiple environments. Some works make parametric assumptions in this case, such as Huang et al. [2020], assuming linearity with non-Gaussian noise. Since we want to avoid strong parametric assumptions, we consider approaches that avoid these assumptions. The principled *Joint Causal Inference* (JCI) framework [Mooij et al., 2020] is one such approach. It demonstrates how to apply traditional constraint-based methods for causal discovery [Glymour et al., 2019] with multi-environment data. In the simpler setting with observed variables $(T, Y, E)$ excluding $X$, the JCI framework informs us that $Y \perp\!\!\!\perp_P E \mid T$ is violated in the presence of a hidden confounder $U$ if $E$ is an instrumental variable. But this means, once again, that the treatment effect is fixed across environments as we assume $E$ has no direct effect on $Y$. Variants of this type of test have also been mentioned by others, for instance Athey et al. [2020, Lemma 3] and Dahabreh et al. [2020]. We demonstrate the limitations of using this approach in our experiments, and provide a more in-depth explanation using graph-based arguments in Appendix C. Our contribution is a more general non-parametric test that works even if $E$ is an invalid instrument that can influence any of the other variables.

## 4  Detecting hidden confounding in multi-environment data

Our goal is to detect the presence of hidden confounding between treatment $T$ and outcome $Y$ after having adjusted for some observed confounders $X$. Graphically, this corresponds to detecting the existence of both edges $U \to T$ and $U \to Y$ in the causal DAG $\mathcal{G}$. In this section, we demonstrate testable conditional independencies between the observed variables that are *only* violated when both those edges exist – hence providing testable implications for hidden confounding.

While we do not assume to know the complete causal DAG $\mathcal{G}$ between our variables, we put two restrictions on it: (i) that $Y$ is not an ancestor of $T$ and (ii) that $X$ is a confounder to both $T$ and $Y$ in contrast to, for instance, being a mediator or only a cause to either one of them. These restrictions are relatively weak as (i) holds in all practical causal inference settings as a treatment $T$ happens before outcome $Y$ in time and (ii) can sometimes be verified by checking that both $T$ and $Y$ depend on $X$. Under this setting, we prove the following.

**Theorem 1.** *Let $\mathbf{T}^{(k)} = (T_1^{(k)}, \ldots, T_{N_k}^{(k)})$ be the vector of all observed treatments in environments $E^{(k)}$; define $\mathbf{Y}^{(k)}$, $\mathbf{X}^{(k)}$, and $\mathbf{U}^{(k)}$ similarly. We consider the data distribution $P(\mathbf{T}^{(k)}, \mathbf{Y}^{(k)}, \mathbf{X}^{(k)}, \mathbf{U}^{(k)})$ with $N_k \geq 2$ under assumption 1,2, 3 and 4. Furthermore, assume an underlying causal DAG $\mathcal{G}$ where $Y$ is not an ancestor of $T$, and that $X$ is a known common cause to $T$ and $Y$. Then, for any $k = 1, \ldots, K$, there exists hidden confounding between $T$ and $Y$ in $\mathcal{G}$ if and only if*

$$T_j^{(k)} \not\!\perp\!\!\!\perp_P Y_i^{(k)} \mid T_i^{(k)}, X_i^{(k)}, X_j^{(k)} \quad \forall i, j = 1, \ldots, N_k : i \neq j . \tag{3}$$

*Proof sketch.* To prove the statement, we look at d-separations in the extended causal graphical model $\mathcal{G}^*$ and show that (3) only is true for corresponding graphs $\mathcal{G}$ where the unobserved $U$ is a confounder between $T$ and $Y$. Figure 1b illustrates how open paths may exist between pairs of observations $(i, j)$ going through $\Theta_T^{(k)}, \Theta_Y^{(k)}, \Theta_X^{(k)}$ or $\Theta_U^{(k)}$ by unrolling the augmented graph $\mathcal{G}^*$. These paths are open because of Assumption 4. This technique resembles the twin network method used for counterfactual inference [Balke and Pearl, 1994] but the results we obtain from using this approach are distinctly different. The complete proof can be found in the Appendix. $\qquad\square$

The variables $T_j^{(k)}$ and $Y_i^{(k)}$ are the treatment and outcome of two different observations in the same environment. Intuitively, the theorem states that after having adjusted for $(T_i^{(k)}, X_i^{(k)}, X_j^{(k)})$, we would expect under the ICM principle that $T_j^{(k)}$ to not provide any information about how $Y_i^{(k)}$ behaves. Thus, if it still does, then this can only be due to unobserved confounding. Testing this independence hence provides us with a testable implication in our observed data distribution on whether the unobserved $U$ is a confounder or not.

**Two-variable case without observed confounders** We can drop the observed confounder $X$ in Theorem 1 and, interestingly, in that case, obtain even stronger results for detecting the presence of a hidden confounder. This setting is interesting as even the two-variable case is notoriously difficult in causal discovery [Reichenbach, 1956, Peters et al., 2017]. Unlike in the more general setting, we no longer need to know the direction of the causal relationship between $T$ and $Y$.

**Theorem 2.** *Let $\mathbf{T}^{(k)} = (T_1^{(k)}, \ldots, T_{N_k}^{(k)})$ be the vector of all observed treatments in environments $E^{(k)}$; define $\mathbf{Y}^{(k)}$ and $\mathbf{U}^{(k)}$ similarly. We consider the data distribution $P(\mathbf{T}^{(k)}, \mathbf{Y}^{(k)}, \mathbf{U}^{(k)})$ without any observed confounders and $N_k \geq 2$ under assumption 1,2, 3 and 4. Then, for any $k = 1, \ldots, K$, there exists hidden confounding between $T$ and $Y$ in $\mathcal{G}$ if and only if*

$$(i) \ \ T_j^{(k)} \not\!\perp\!\!\!\perp_P Y_i^{(k)} \mid T_i^{(k)} \quad and \quad (ii) \ \ T_j^{(k)} \not\!\perp\!\!\!\perp_P Y_i^{(k)} \mid Y_j^{(k)} \quad \forall i, j = 1, \ldots, N_k : i \neq j . \tag{4}$$

Guo et al. [2022] studied a similar setting to Theorem 2 and demonstrated how to decide the causal direction between $T$ and $Y$ in this case when there is no latent variable. Our results extend theirs as we now also show how to exclude the possibility of a latent common cause in this setting. The proof is similar to that of Theorem 1, but the conditional independencies are different. Firstly, we have $T_j^{(k)} \not\!\perp\!\!\!\perp_P Y_i^{(k)} \mid T_i^{(k)}$ which is the conditional independence in Theorem 1 without conditioning on $X_i^{(k)}$ and $X_j^{(k)}$. Secondly, we have $T_j^{(k)} \not\!\perp\!\!\!\perp_P Y_i^{(k)} \mid Y_j^{(k)}$. This one is necessary as we no longer assume anything about the ancestral relationship between treatment and outcome. If we had assumed that $T$ could not be a descendant of $Y$, we can show that only condition (i) in the theorem is necessary. Similarly, condition (ii) is only necessary when $Y$ could not be a descendant of $T$.

## 4.1 Influence of the assumptions

Our theory shows how to test for hidden confounding, but it now relies on other untestable assumptions: namely non-degenerate independent causal mechanisms and the faithfulness & causal Markov property. Due to this, we investigate the necessity of these assumptions and identify various failure cases when they are violated. On a more positive note, we also demonstrate that the assumption of non-degenerate mechanisms can be weakened. We present here the main conclusions regarding violations on two of the assumptions while more elaborate explanations can be found in Appendix D, together with a demonstration of how our procedure can fail due to faithfulness violations as well as a discussion on assumptions about positivity and selection bias.

**Violation of Assumption 3: dependent causal mechanisms**  What happens if any of the pair-wise independencies between $\Theta_T, \Theta_Y, \Theta_X$ or $\Theta_U$ are violated? To investigate this, we go through the same procedure for proving Theorem 1 where we allow any of these mechanisms to be dependent. We find that $T_j^{(k)} \perp\!\!\!\perp_P Y_i^{(k)} \mid T_i^{(k)}, X_i^{(k)}, X_j^{(k)}$ can be violated even when there is no confounding in all but one case with dependent mechanisms, meaning that it no longer works for detecting hidden confounding. The only case where our theory still works is when $\Theta_X \not\!\perp\!\!\!\perp_P \Theta_U$ – i.e. the mechanisms of the observed and unobserved confounders are allowed to co-vary across environments.

**Violation of Assumption 4: degenerate causal mechanisms**  What happens if one or more of the distributions $P(\Theta_T), P(\Theta_Y), P(\Theta_X)$ and $P(\Theta_U)$ are degenerate, meaning that some mechanisms are fixed across all environments? In the most extreme case, if all mechanisms are fixed then the distribution $P_E$ would be identical in each environment. We investigate these scenarios by first adding $\Theta_T, \Theta_Y, \Theta_X$ and/or $\Theta_U$ to the conditioning set of the independence in Theorem 1. Then, we check whether this independence still is violated in the presence of hidden confounding using the same procedure used for proving the theorem. We find that the theorem fails only when we condition on both $\Theta_T$ and $\Theta_U$. In other words, it is only strictly necessary for our theory that changes in $P_{\Theta_T}(T \mid \mathrm{Pa}(T))$ or $P_{\Theta_U}(U \mid \mathrm{Pa}(U))$ occur between environments.

**Remark 1.** *We may now identify a more conservative interpretation of our proposed procedure. First, one can verify the assumption of non-degenerate causal mechanisms by checking from data whether $P_E(T \mid X)$ varies across environments; if it does, then that is likely because $\Theta_T$ and/or – through potential downstream effects – $\Theta_U$ are non-degenerate. Next, we would run our proposed procedure. Now if the null is rejected then we can be conservative by concluding that this is either because we have hidden confounding and/or dependent mechanisms. But in the case of no rejection, it can only be interpreted as having no hidden confounders present. This is because having dependent causal mechanisms (violation of assumption 3) can only cause false positives.*

## 4.2 Testing the independence

Here, we explain how test the conditional independence $T_j^{(k)} \perp\!\!\!\perp_P Y_i^{(k)} \mid T_i^{(k)}, X_i^{(k)}, X_j^{(k)}$ from Theorem 1 using multi-environment data; the full procedure is summarized in Algorithm 1 where we have defined $t_{2i-1}^{\pi_{2i}} := \{t_{2i-1}^{(k)}\}_{k \in \pi_{2i}}$ and similarly for $y_{2i}^{\pi_{2i}}, t_{2i}^{\pi_{2i}}, x_{2i-1}^{\pi_{2i}}$ and $x_{2i}^{\pi_{2i}}$.

To test $T_j^{(k)} \perp\!\!\!\perp_P Y_i^{(k)} \mid T_i^{(k)}, X_i^{(k)}, X_j^{(k)}$, we need to simulate sampling from the joint distribution $P(T_i^{(k)}, Y_i^{(k)}, X_i^{(k)}, T_j^{(k)}, Y_j^{(k)}, X_j^{(k)})$. Note here that we do not condition on $E^{(k)}$. The idea is as follows: we select two different observations $i$ and $j$ from all environments such that we get a vector of observed treatments $t_i = (t_i^{(1)}, t_i^{(2)}, \ldots, t_i^{(K)})$; outcomes $y_i = (y_i^{(1)}, y_i^{(2)}, \ldots, y_i^{(K)})$; and so on for $x_i, t_j$ and $x_j$. Then, we can use any suitable method for conditional independence testing with $t_i, y_i, x_i, t_j$ and $x_j$. Note that the choice of observations within each environment is arbitrary as long as we do not pick the same observation for $i$ and $j$, this is a consequence of observations being i.i.d. within each environment.

**Increasing power of test with Fisher's method**  In essence, we perform a conditional independence test where the "sample size" of the test is the number of available environments. Thus, for a small

**Algorithm 1:** Algorithm for statistically testing the presence of hidden confounding

---

**Input:** Datasets $D_k = \{(t_i^{(k)}, y_i^{(k)}, x_i^{(k)})\}_{i=1}^{N_k}$ from environments $k = 1, \ldots, K$; significance level $\alpha$; minimum number of environments required for hypothesis test $K_{\min}$

1   $L_{\max} \leftarrow \text{ceiling}(\max_k N_k/2)$          The maximum number of possible hypothesis tests

2   **for** $i = 1, \ldots, L_{\max}$ **do**

3      $\pi_{2i} \leftarrow \{k \in [K] : N_k \leq 2i\}$        Retrieve all environments with at least $2i$ samples

4      **if** $\text{size}(\pi_{2i}) < K_{\min}$ **then**

5          $L_{\max} \leftarrow i - 1$

6          break          Stop for-loop if we run out of a sufficient number of environments

7      $p_i \leftarrow \text{cond\_indep\_test} \, (t_{2i-1}^{\pi_{2i}} \perp\!\!\!\perp y_{2i}^{\pi_{2i}} \mid t_{2i}^{\pi_{2i}}, x_{2i-1}^{\pi_{2i}}, x_{2i}^{\pi_{2i}})$ Get p-value from independence test

8   $z \leftarrow -2 \sum_{i=1}^{L_{\max}} \log(p_i)$          Aggregate p-values with Fisher's method

9   $p \leftarrow 1 - \text{cdf}_{\chi^2_{2L_{\max}}}(z)$          Compute global p-value

10   **return** $p \leq \alpha$

---

number of environments, our test might have low power (probability of detecting hidden confounding when it is present). To alleviate this issue, we recognize that we can perform this test multiple times if we have many samples $N_k$ per environment. Then, we select new observations from every environment for each hypothesis test until all observations have been used up. It is important to note that we only select from environments where there still are observations that have not yet been used for the hypothesis testing. Since each hypothesis test is independent and has the same null, we can aggregate the p-values from all tests using Fisher's method to obtain a global hypothesis test [Fisher, 1925]. As we show in our experiments, using Fisher's method drastically improves the power of our method, thus reducing the number of environments needed to detect the presence of hidden confounding. Having a different number of samples per environment $N_k$ also necessitates specifying the hyperparameter $K_{\min}$, which determines the minimum observations required in each hypothesis test. This parameter should be chosen to ensure that the used independent testing method works properly if it is provided with at least $K_{\min}$ samples.

## 5   Experiments

To evaluate and investigate the theory for testing hidden confounding in multi-environment data, we perform a series of simulation studies with synthetic data in addition to experiments with semi-synthetic data generated using the Twins dataset [Almond et al., 2005, Louizos et al., 2017].[2] As we want to evaluate our method's ability to detect confounding, we use data where the ground-truth causal graph is known. Unless otherwise stated, each experiment is repeated 50 times where we use a significance level $\alpha = 0.05$. Depending on the variable types in the experiment, we state what suitable conditional independence testing method is used by our algorithm.

### 5.1   Synthetic data

For the synthetic data experiments, we generate data as follows: we have the confounder $U_i^{(k)} \sim \text{Normal}(\Theta_U^{(k)}, 1)$; treatment $T_i^{(k)} \sim \text{Ber}(\text{Sigm}(U_i^{(k)} + \Theta_T^{(k)}))$; and outcome $Y_i^{(k)} \sim \text{Ber}(\text{Sigm}(\lambda U_i^{(k)} + T_i^{(k)} + \Theta_Y^{(k)}))$ . Note $\text{Sigm}(x) = 1/(1 + e^{-x})$ is the logistic function and $\Theta_V^{(k)} \sim \text{Normal}(0, \sigma^2_{\Theta_V})$ for $V \in \{T, Y, U\}$. Unless otherwise stated, we use $\sigma_{\Theta_T} = \sigma_{\Theta_U} = \sigma_{\Theta_Y} = 1$. We control the strength of confounding by varying $\lambda$, where $\lambda = 0$ corresponds to no confounding.

---

[2]Code available at github.com/RickardKarl/detect-hidden-confounding.

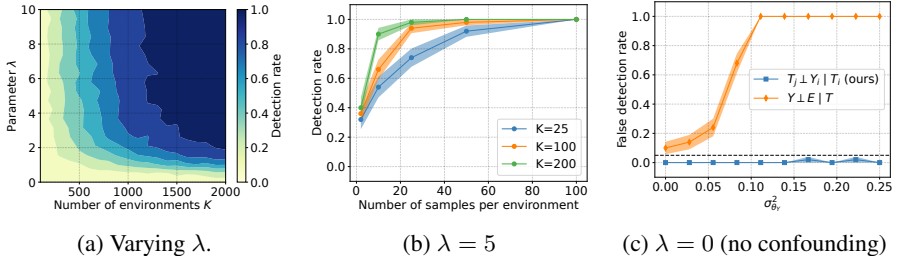

|  (a) Varying $\lambda$. | (b) $\lambda = 5$ | (c) $\lambda = 0$ (no confounding) |

Figure 3: **Synthetic data** – **(a)**: Detecting confounding with $N_k = 2$ across a range of confounder effect sizes and numbers of environments $K$. (500 repetitions) **(b)**: Simulations with fixed confounding strength $\lambda = 5$ for $N_k > 2$ with a small number of environments $K$. **(c)**: Comparing the proposed procedure and an alternative testing procedure by varying the standard deviation of $\Theta_Y$ in the absence of confounding. The black dashed line corresponds to the desired type 1 error control $\alpha = 0.05$. The shaded area shows the standard error from 50 repetitions.

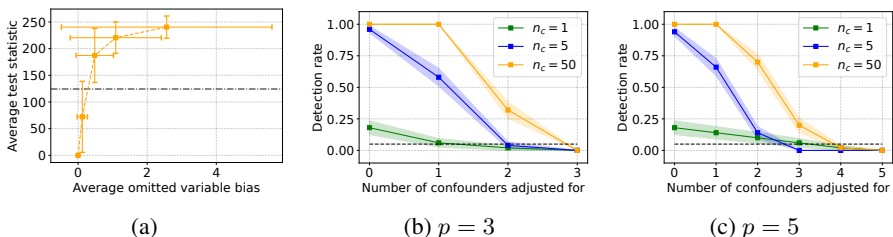

|  (a)  | (b) $p = 3$ | (c) $p = 5$ |

Figure 4: **Twins dataset** – **(a)**: Effect of bias from omitting confounders on the test statistic of our hypothesis test. **(b)**, **(c)**: Performance when adjusting for observed confounders with either a total of 3 or 5 confounders in the data. The different curves correspond to combining numbers of hypothesis tests. The black dashed line corresponds to the rejection threshold / desired type 1 error $\alpha = 0.05$ in all figures, and the error bars / shaded area shows standard deviation (figure a) or standard error (figures b and c) from 50 repetitions.

**Larger confounder effect sizes increase the probability of detection**   We investigate how the effect size of the confounding variable influences our proposed testing procedure. We vary $\lambda$ between 0 (no confounding) and 10 while also varying the number of environments. We perform this experiment with $N_k = 2$ for all $k$ and use the G-test for conditional independence testing [McDonald, 2014]. The results are shown in Figure 3a. We note two things: the probability of detection grows for larger confounder effect size and it also grows when the number of environments is increased.

**The growth rate in detection depends on the number of environments**   We investigate the probability of detecting confounding when varying both the number of environments and the number of samples per environment for a fixed confounding strength $\lambda = 5$. We use a permutation-based method for the conditional independence test [Tsamardinos and Borboudakis, 2010] as we do not want to rely only on asymptotic validity (such as in the G-test) due to the limited number of environments. The results show that the performance of the testing procedure is highly dependent on the number of environments $K$, see 3b. The probability of detection grows as we increase the number of samples. Noticeably, the rate of growth increases with the number of environments $K$.

**Robustness to environmental changes**   We compare our proposed procedure to the alternative approach of testing $Y \perp\!\!\!\perp_P E \mid T$ to detect hidden confounding, the latter being valid when $E$ is an instrumental variable [Mooij et al., 2020]. Here we test the sensitivity to violating one of its conditions, namely that $P_E(Y \mid T)$ is fixed under the null. We vary $\sigma_{\Theta_Y}$ between 0 and $\frac{1}{4}$ when there is no confounding by setting $\lambda = 0$ with $N = 100$ and $K = 500$, and we use the G-test for conditional independence testing [McDonald, 2014]. As shown in Figure 3c, the probability of false detection using $Y \perp\!\!\!\perp_P E \mid T$ increases when $\sigma_{\Theta_Y}$ starts to increase. Meanwhile, the false detection rate (type 1 error) remains bounded by $\alpha = 0.05$ for our procedure as desired. In Appendix F, we

also include the same comparison when confounding is present to confirm that our method is able to detect confounding in this case.

## 5.2 Twins dataset

We use data from twin births in the USA between 1989-1991 [Almond et al., 2005, Louizos et al., 2017] to construct an observational dataset with continuous treatment/outcome and non-linear relationships. Here the environments are different states, and a notable element of our dataset is that all variations between environments stem solely from the real-world distribution shifts of the covariates between birth states. The strength of confounding is controlled by a parameter $\lambda$, where $\lambda = 0$ corresponds to no confounding. The full procedure for data generation is described in Appendix E. For the following experiments, we use the Kernel Conditional Independence Test [Zhang et al., 2012] in our algorithm due to having continuous variables and, unless otherwise stated, combine 50 hypothesis tests using Fisher's method.

**Detection rate increases with bias from unobserved confounding**    We perform an experiment having $p = 5$ unobserved confounders, where we vary confounding strength $\lambda$ between 0 and 5. We compute the bias from omitting the unobserved confounders when estimating the average treatment effect of $T$ on $Y$ in each environment. We then compare the average bias to the test statistic computed by our algorithm averaged over multiple iterations. As observed in Figure 4a, the test statistic increases together with the bias. The black dashed line in the figure represents the rejection threshold at $\alpha = 0.05$, hence we can see that for sufficient bias the method will detect it.

**Adjusting for observed confounders**    In the last experiments, we attempt to detect hidden confounding while also adjusting for observed confounders. We go from observing none to all confounders while having a confounding strength of $\lambda = 5$. We do this for the case with either a total of $p = 3$ or $p = 5$ confounders, shown in Figure 4b and 4c, respectively. In addition, we investigate the influence of combining multiple hypothesis tests ($n_c$ denotes the number of tests) using Fisher's method. We observe first that adjusting for more confounders leads to a decrease in detection rate, and that our desired type 1 error of $\alpha = 0.05$ is controlled when we have adjusted for all confounders. Secondly, the performance deteriorates when the total number of confounders increases, as indicated by the detection rate, which is lower when adjusting for 4 confounders when $p = 5$ than adjusting for 2 confounders when $p = 3$. This is likely because the conditional independence test loses power as the conditioning set becomes larger [Zhang et al., 2012]. Thirdly, we see that the combination of multiple hypothesis tests using Fisher's method does improve the power of our algorithm. We did, however, not see any significant benefit in combining more than 50 hypothesis tests in these experiments.

## 6    Discussion

In this work, we studied a setting where observational data has been collected from different heterogeneous environments in which the same treatment $T$, outcome $Y$, and covariates $X$ have been observed. We showed that assuming independent causal mechanisms, there exist testable conditional independencies that are violated in the presence of hidden confounders, for which we also proposed a statistical procedure to test these independencies from observed data. In many cases, with a sufficient number of environments, we show that we are able to detect confounding when it is present. While our main goal was to derive testable implications of hidden confounding, open questions remain on how to improve sample efficiency and tackle loss of power when adjusting for many observed confounders. Addressing these can lead to better tools for researchers to validate their causal assumptions and move towards making safer causal inferences.

**Societal impact**    Causal inference has a big influence on real-world decision-making as it lies at the core of many sciences, ranging from medicine to public policy. While our work has the potential to improve the soundness and safety of causal inference methodology, this research is still in its infancy and we caution careful use of this work, particularly in high-stakes settings.

## Acknowledgements

The authors thank Marco Loog, Stephan Bongers, Alexander Mey, and Frans Oliehoek, and the members of the Pattern Recognition lab for their invaluable feedback on earlier versions of this manuscript. We also thank our anonymous reviewers for their helpful comments and input.

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

## Appendix

The appendix contains the following sections.

# A Future Work

We have identified a set of open questions deriving from our work. Firstly, our theory applies to the common setting in causal inference with treatment $T$, outcome $Y$, and a possibly high-dimensional confounder $X$, in which we want to detect the presence of additional hidden confounding $U$ (that also can be high-dimensional). While this is arguably the most typical setting in causal inference, it is of interest to consider other scenarios with more variables and interactions between them. We conjecture that other testable implications exist for confounding in these settings that could be found with similar arguments as we use. A particularly interesting setting is when we observe a proxy to a hidden confounder, which can be used for adjustment instead [Kuroki and Pearl, 2014, Miao et al., 2018]. In this case, it is no longer straightforward to say whether there could be a hidden confounder that is unrelated to the proxy. We also believe that our techniques might be applicable to scenarios with instrumental variables (IV) [Angrist et al., 1996, Martens et al., 2006], to test whether there exists any confounding between the IV and treatment which is a requirement for valid IV estimation.

Secondly, our theorem fundamentally relies on a set of untestable assumptions: independent & non-degenerate causal mechanisms and the faithfulness & causal Markov property. Although we investigated various violations, these results raised new questions. In particular, the effect of faithfulness violations, perhaps surprisingly, had a large influence on our procedure. Therefore, it is important to understand whether similar observations can be made in more realistic settings.

Thirdly, an interesting direction for future research would be to investigate how our approach can be used to estimate confounding strength, to be used in well-studied approaches in sensitivity analysis [Rosenbaum and Rubin, 1983b, Cinelli et al., 2019].

Lastly, while our main goal was to derive testable implications of hidden confounding, there are opportunities to improve the way we test these from data. We observed that our proposed procedure sometimes requires a large number of environments. While it is unclear whether this is a property of the theory or the lack of efficiency in the test procedure we used, we note that combining multiple hypothesis tests using Fisher's method helped with performance. A possible research direction could be to investigate how to refine this approach further. Further, we observed performance deteriorating as we adjusted for more observed confounders, likely due to the curse of dimensionality [Zhang et al., 2012]. A promising solution here could be to use popular dimensionality-reduction techniques from causal inference such as the propensity score [Rosenbaum and Rubin, 1983a].

# B Proofs

In this section, we present the proof for Theorem 1 and 2. Let $\mathbf{T}^{(k)} = (T_1^{(k)}, \ldots, T_{N_k}^{(k)})$ be the vector of all observed treatments in environments $E^{(k)}$. Define $\mathbf{Y}^{(k)}$, $\mathbf{X}^{(k)}$, and $\mathbf{U}^{(k)}$ similarly.

**Theorem 1.** *We consider the data distribution $P(\mathbf{T}^{(k)}, \mathbf{Y}^{(k)}, \mathbf{X}^{(k)}, \mathbf{U}^{(k)})$ with $N_k \geq 2$ under assumption 1,2, 3 and 4. Furthermore, assume an underlying causal DAG $\mathcal{G}$ where $Y$ is not an ancestor of $T$, and that $X$ is a known common cause to $T$ and $Y$. Then, for any $k = 1, \ldots, K$, there exists hidden confounding between $T$ and $Y$ in $\mathcal{G}$ if and only if*

$$T_j^{(k)} \not\perp\!\!\!\perp_P Y_i^{(k)} \mid T_i^{(k)}, X_i^{(k)}, X_j^{(k)} \quad \forall i, j = 1, \ldots, N_k : i \neq j \,. \tag{5}$$

*Proof.* We constrain ourselves to DAGs $\mathcal{G}$ with variables $(T, Y, XU)$ where $X$ is a known common cause to both $T$ and $Y$, and $Y$ is not an ancestor of $T$ in $\mathcal{G}$. Under the assumption of non-degenerate, independent causal mechanisms (Assumption 3 and 4), we can introduce the mechanisms $\Theta_V$ for each variable $V \in \{T, Y, X, U\}$. Further, we can augment $\mathcal{G}$ with a hierarchical structure [Gelman et al., 2013, Chapter 5], wherein we first sample the mechanisms i.i.d. $\Theta_V^{(k)} \sim P(\Theta_V)$ for $k = 1, \ldots, K$ and $V \in \{T, Y, X, U\}$ and then, for each environment $k$, obtain $(T_i^{(k)}, Y_i^{(k)}, X_i^{(k)}, U_i^{(k)})$ by repeatedly sampling $N_k$ times conditioned on the mechanisms. We denote this augmented graph with the hierachical structure as $\mathcal{G}^*$, where edges between $(T_i^{(k)}, Y_i^{(k)}, X_i^{(k)}, U_i^{(k)})$ are the same as for $(T, Y, X, U)$ and $\Theta_V^{(k)} \in \mathrm{Pa}(V_i^{(k)})$ where $V \in \{T, Y, X, U\}$, for all $i$ and $k$. An example of

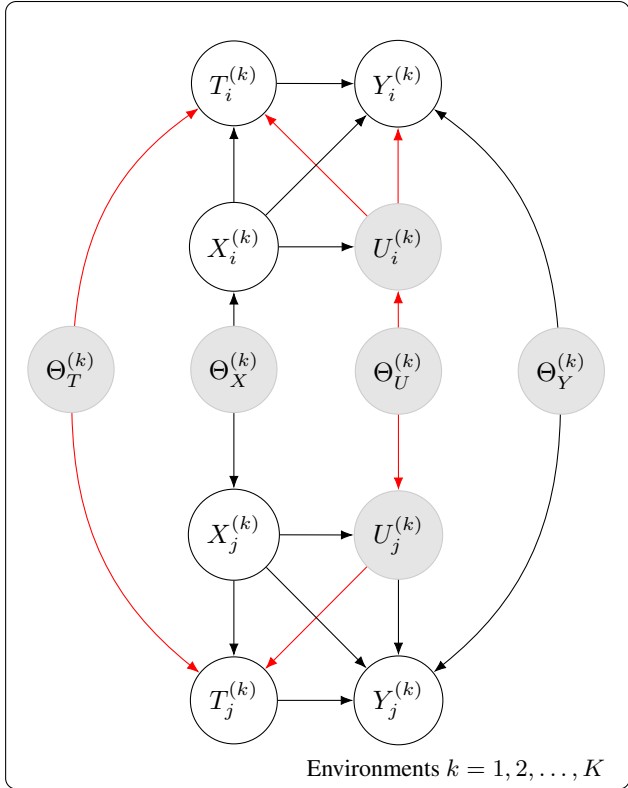

Figure 5: Example of unrolling the augmented causal DAG $\mathcal{G}^*$ for some pair of observations $(i,j)$. The samples are generally not independent due to the shared mechanisms $(\Theta_T, \Theta_X, \Theta_U, \Theta_Y)$, unless we condition on them. Confounding is present, and the red edges mark open paths such that $T_j^{(k)} \not\perp\!\!\!\perp_d Y_i^{(k)} \mid T_i^{(k)}, X_i^{(k)}, X_j^{(k)}$ in this graph. Note that these paths go through the outgoing edges from $U_i^{(k)}$ to $T_i^{(k)}$ and $Y_i^{(k)}$ (and similarly for $j$), and that the paths are closed if either of the edges is removed.

such an augmented graph $\mathcal{G}^*$ is shown in Figure 5. Notably, this augmentation can be done for all $k$ because we assume that all environments share the same causal graph $\mathcal{G}$ (Asssumption 2).

Now, given the constraints that we have defined, we consider every combination of the edges between $(T, Y, X, U)$ that are DAGs. In total, there are 40 different DAGs that encompass all these combinations of edges. We say that $U$ is a confounder in one of these DAGs if both the edges $U \to T$ and $U \to Y$ exist. For each of these graphs $\mathcal{G}$, we shall investigate the d-separations in its augmented version $\mathcal{G}^*$. Notably, due to the assumption of non-degenerate mechanisms (Assumption 4), we allow open paths in $\mathcal{G}^*$ that go through $\Theta_T^{(k)}, \Theta_Y^{(k)}, \Theta_X^{(k)}$, or $\Theta_U^{(k)}$. This means that two different observations $\left(T_i^{(k)}, Y_i^{(k)}, X_i^{(k)}, U_i^{(k)}\right)$ and $\left(T_j^{(k)}, Y_j^{(k)}, X_j^{(k)}, U_j^{(k)}\right)$ can be dependent when $i \neq j$ for $i, j = 1, \ldots, N_k$. These paths are best illustrated by unrolling the augmented graph $\mathcal{G}^*$ as in the example in Figure 5. However, such dependencies can only happen if we do not condition on the mechanisms (that is, the environment) as we know that the observations are sampled i.i.d. within each environment. This demonstrates the need for multiple environments, as the randomness from sampling $\left(\Theta_T^{(k)}, \Theta_Y^{(k)}, \Theta_X^{(k)}, \Theta_U^{(k)}\right)$ allows us to treat them as ordinary random variables. To capture this randomness, we need to observe multiple environments though. Note that we also need $N_k \geq 2$ for $i \neq j$ to hold.

We will pay attention to a particular d-separation in $\mathcal{G}^*$, namely

$$T_j^{(k)} \perp\!\!\!\perp_d Y_i^{(k)} \mid T_i^{(k)}, X_i^{(k)}, X_j^{(k)} \ .$$

We automatically iterate over our list of DAGs using the *dagitty* package in R [Textor et al., 2017] and check whether this d-separation holds; the results are displayed in Table 1. We note that the shaded rows in the table are the cases where $U$ is a confounder, and these are the only cases where $T_j^{(k)} \perp\!\!\!\perp_d Y_i^{(k)} \mid T_i^{(k)}, X_i^{(k)}, X_j^{(k)}$ is violated in $\mathcal{G}^*$. In other words, checking this d-separation is sufficient to determine whether $U$ is a confounder in $\mathcal{G}$. Assuming the faithfulness and causal Markov property (Assumption 1), we have that:

$$T_j^{(k)} \not\!\perp\!\!\!\perp_d Y_i^{(k)} \mid T_i^{(k)}, X_i^{(k)}, X_j^{(k)} \iff T_j^{(k)} \not\!\perp\!\!\!\perp_P Y_i^{(k)} \mid T_i^{(k)}, X_i^{(k)}, X_j^{(k)} .$$

Consequently, it follows that

$$T_j^{(k)} \not\!\perp\!\!\!\perp_P Y_i^{(k)} \mid T_i^{(k)}, X_i^{(k)}, X_j^{(k)} \quad \text{for } i \neq j \iff \text{U is a confounder to } T \text{ and } Y .$$

This result holds for any $k$ since we assume that all environments share the same causal DAG (Assumption 2).

$\square$

**Remark 2.** *$X_j^{(k)}$ can be removed from the conditioning set in the independence of Theorem 1 when we assume that the observed and unobserved confounders are independent of each other. In practice, however, we most likely would not like to make this assumption which is why we recommend to condition on both $X_i^{(k)}$ and $X_j^{(k)}$.*

**Theorem 2.** *We consider the data distribution $P(\mathbf{T}^{(k)}, \mathbf{Y}^{(k)}, \mathbf{U}^{(k)})$ without any observed confounders and $N_k \geq 2$ under assumption 1,2, 3 and 4. Then, for any $k = 1, \ldots, K$, there exists hidden confounding between $T$ and $Y$ in $\mathcal{G}$ if and only if*

$$(i) \ \ T_j^{(k)} \not\!\perp\!\!\!\perp_P Y_i^{(k)} \mid T_i^{(k)} \quad and \quad (ii) \ \ T_j^{(k)} \not\!\perp\!\!\!\perp_P Y_i^{(k)} \mid Y_j^{(k)} \ \ \forall i, j = 1, \ldots, N_k : i \neq j . \quad (6)$$

*Proof.* Using the same arguments as in the proof of Theorem 1, we can show that $T_j^{(k)} \not\!\perp\!\!\!\perp_P Y_i^{(k)} \mid T_i^{(k)}$ and $T_j^{(k)} \not\!\perp\!\!\!\perp_P Y_i^{(k)} \mid Y_j^{(k)}$ exclusively holds for those where there exists no common cause between $T$ and $Y$ when we only have the variables $(T, Y, U)$, excluding any observed confounder $X$. The corresponding table to check this is demonstrated in Table 2. $\square$

## C Alternative Test for Detecting Hidden Confounding in Two-variable Case

In this section, we explain when (and when not) testing $Y \perp\!\!\!\perp_P E \mid T$ is a valid strategy for detecting hidden confounding between $T$ and $Y$. We require that $Y \perp\!\!\!\perp_P E \mid T, U$ when there is no confounding meaning that there can not exist any direct causal relationships $E \to Y$ or $E \to U$; if this does not hold then testing $Y \perp\!\!\!\perp_P E \mid T$ would not be informative. This is demonstrated with the dashed edges in Figure 6a. Meanwhile, in Figure 6b, we display the case where a violation of $Y \perp\!\!\!\perp_P E \mid T$ correctly detects hidden confounding being present. The conclusion is that this alternative test, which is easy to test, only works if the changes between environments happen for the conditional distribution $P_E(T \mid \text{Pa}(T))$, as only $T$ may be directly caused by $E$ if this test shall work. This corresponds to $E$ being an instrumental variable [Angrist et al., 1996], meaning that $T \not\!\perp\!\!\!\perp_P E$, $U \perp\!\!\!\perp_P E$, and the environment $E$ can not have any direct effect on the outcome $Y$ under the null (that is no confounding).

## D Further Analysis on the Influence of the Assumptions

In this section, we present more elaboration on the examples of violating the assumptions behind our theory, as discussed in Section 4.1. First, we provide a counter-example that shows how Theorem 1 fails when we have dependent causal mechanisms. Secondly, we demonstrate how we got to our conclusions on having degenerate mechanisms. Thirdly, we provide insights into when faithfulness violations could occur in an example with a linear-Gaussian structural causal model. Finally, we discuss the analogy to the positivity assumption for our theory as well as how our procedure could be influenced by the presence of selection bias.

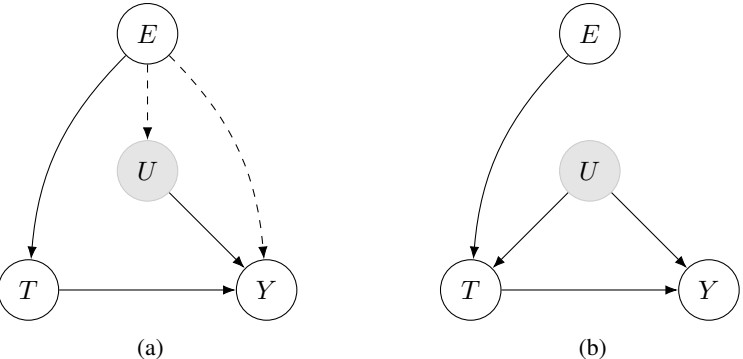

Figure 6: Examples of using $Y \perp\!\!\!\perp_P E \mid T$ to detect confounding. Gray corresponds to unobserved variables. **(a)** Violations of $Y \perp\!\!\!\perp_P E \mid T$ despite there being no confounding between $T$ and $Y$. Removing the two dashed edges would make $Y \perp\!\!\!\perp_P E \mid T$ hold. **(b)** Case with correct violation of $Y \perp\!\!\!\perp_P E \mid T$ when unobserved confounding is present.

## D.1 Dependent Mechanisms

We wish to demonstrate examples of DAGs where the main condition

$$T_j^{(k)} \perp\!\!\!\perp_P Y_i^{(k)} \mid T_i^{(k)}, X_i^{(k)}, X_j^{(k)}$$

in Theorem 1 is violated despite that there is no confounding when some of the causal mechanisms are pairwise dependent. We are able to find examples of violations with all pairwise dependencies, except for $\Theta_X \not\perp\!\!\!\perp_P \Theta_U$. Hence, we conjecture that a dependency between the mechanism of the observed and unobserved confounder does not influence the test proposed by our theory. Figures 7a, 7b, 7c, 8a and 8b show found examples where violations occur between the mechanisms.

## D.2 Degenerate Mechanisms

What happens if one or more of the distributions $P(\Theta_T)$, $P(\Theta_Y)$, $P(\Theta_X)$, and $P(\Theta_U)$ are constant across all environments? We investigate these scenarios by first adding $\Theta_T$, $\Theta_Y$, $\Theta_X$ and/or $\Theta_U$ to the conditioning set in $T_j^{(k)} \perp\!\!\!\perp_P Y_i^{(k)} \mid T_i^{(k)}, X_i^{(k)}, X_j^{(k)}$ - the testable implication for confounding. We then follow the same procedure as for proving Theorem 1 and noting whether this conditional independence still discriminates between cases when $U$ is and is not a confounder. Out of 15 possible cases with degenerate mechanisms, we identify that the theorem fails every time we condition on both $\Theta_T$ and $\Theta_U$, as demonstrated in Table 3.

## D.3 Faithfulness Violation

What happens if the conditional independencies we test do not correspond to the dependencies in the underlying causal graph? If that is the case, we would fail to detect dependencies implying the presence of confounders. Knowing when a faithfulness violation occurs is not possible, we can only reason about its plausibility. In this section, we first showcase an example where a faithfulness violation occurs in linear-Gaussian structural causal models for particular configurations as well as when the confounder effect sizes – that is $\gamma$ and $\lambda$ in (7) – become very large.

It was proved by Meek [1995] that all graphs with discrete and linear-Gaussian data distributions fulfill faithfulness in a measure-theoretic sense; distributions that violate faithfulness have measure zero.

However, even if we restrict $T$ and $Y$ to be categorical we might not want to assume that $U$ or the causal mechanisms follow a discrete distribution. The following example also demonstrates the practical issues stemming from faithfulness violations even when the data is jointly Gaussian.

**Example 1.** *Let $k = 1, \ldots, K$ and $i = 1, \ldots, N_k$. Consider the structural causal model*

$$
\begin{aligned}
U_i^{(k)} &= \Theta_U^{(k)} + \varepsilon_{U,i}, & \varepsilon_{U,i} &\sim \text{Normal}(0, \sigma_U^2), \\
T_i^{(k)} &= \gamma U_i^{(k)} + \Theta_T^{(k)} + \varepsilon_{T,i}, & \varepsilon_{T,i} &\sim \text{Normal}(0, \sigma_T^2), \\
Y_i^{(k)} &= \lambda U_i^{(k)} + \beta T_i^{(k)} + \Theta_Y^{(k)} + \varepsilon_{Y,i}, & \varepsilon_{Y,i} &\sim \text{Normal}(0, \sigma_Y^2),
\end{aligned}
\tag{7}
$$

*where $\Theta_V^{(k)} \sim \text{Normal}(0, \sigma_{\Theta_V}^2)$ and $\varepsilon_V \perp\!\!\!\perp_P \Theta_V^{(k)}$ for $V \in \{T, Y, U\}$. Further, subscript $i$ for the noise variables indicates that they are sampled independently for each observation $i$. Then, despite the presence of confounding, $T_j^{(k)} \perp\!\!\!\perp_P Y_i^{(k)} \mid T_i^{(k)}$ for any $i \neq j$ when $\sigma_{\Theta_U} = \frac{\sigma_U}{\sigma_T}\sigma_{\Theta_T}$. In the finite-sample setting, it noticeably influences our ability to detect confounding when the distribution parameters come close to this equality, as illustrated in Figure 9.*

To create Figure 9, we generate data according to (7) with the following parameters fixed: $\beta = \gamma = \lambda = 1$, $\sigma_{\Theta_Y} = 1$, $\sigma_Y = \sigma_U = 1$ and $\sigma_T = \frac{2}{3}$. Meanwhile, we vary $\sigma_{\Theta_T}$ and $\sigma_{\Theta_U}$ between 0 and 5. We test $T_j^{(k)} \perp\!\!\!\perp_P Y_i^{(k)} \mid T_i^{(k)}$ using the partial correlation [Baba et al., 2004]. The experiment is repeated 1000 times with 1000 environments and a significance level $\alpha = 0.05$.

### D.4 Derivation of Example 1

We consider the structural causal model in (7). Now, we want to prove that $T_j^{(k)} \perp\!\!\!\perp_P Y_i^{(k)} \mid T_i^{(k)}$ for any $i \neq j$ when $\sigma_{\Theta_U} = \frac{\sigma_U}{\sigma_T}\sigma_{\Theta_T}$. For ease of notation, we will drop the superscript $(k)$, as the results hold for any $k$.

Crucially, we note that the partial correlation

$$
\rho_{T_j, Y_i \cdot T_i} = \frac{\rho_{T_j, Y_i} - \rho_{T_j, T_i}\rho_{T_i, Y_i}}{\sqrt{1 - \rho_{T_j, T_i}^2}\sqrt{1 - \rho_{T_i, Y_i}^2}} ,
\tag{8}
$$

is zero if and only if $T_j \perp\!\!\!\perp_P Y_i \mid T_i$ when the data is jointly Gaussian [Baba et al., 2004], which is the case for (7) because $p(T_i, Y_i, U_i) = P(Y_i \mid T_i, U_i)P(T_i \mid U_i)P(U_i)$ where each factor is a Gaussian density.

To check when the partial correlation is zero, we need to find out when

$$
\rho_{T_j, Y_i} - \rho_{T_j, T_i}\rho_{T_i, Y_i} = 0 .
$$

Since $\rho_{X,Y} = \frac{\text{Cov}(X,Y)}{\sqrt{\text{Var}(X)\text{Var}(Y)}}$ for some random variables $X$ and $Y$, we can write this as

$$
\rho_{T_j, Y_i} - \rho_{T_j, T_i}\rho_{T_i, Y_i} = \frac{\text{Cov}(T_j, Y_i)}{\sqrt{\text{Var}(T_j)\text{Var}(Y_i)}} - \frac{\text{Cov}(T_j, T_i)}{\sqrt{\text{Var}(T_j)\text{Var}(T_i)}}\frac{\text{Cov}(T_i, Y_i)}{\sqrt{\text{Var}(T_i)\text{Var}(Y_i)}}
\tag{9}
$$

$$
= \frac{\text{Var}(T_i)\text{Cov}(T_j, Y_i) - \text{Cov}(T_j, T_i)\text{Cov}(T_i, Y_i)}{\sqrt{\text{Var}(T_i)^3\text{Var}(Y_i)}} ,
\tag{10}
$$

where we used the fact that $\text{Var}(T_i) = \text{Var}(T_j)$ for any samples $i$ and $j$.

First, we need to determine all the (co)variances, for which we need to know

$$
\begin{aligned}
T_i &= \gamma\Theta_U + \gamma\varepsilon_{U,i} + \Theta_T + \varepsilon_{T,i} \\
Y_i &= \lambda\Theta_U + \lambda\varepsilon_{U,i} + \beta\gamma\Theta_U + \beta\gamma\varepsilon_{U,i} + \beta\Theta_T + \beta\varepsilon_{T,i} + \Theta_Y + \varepsilon_{Y,i}
\end{aligned}
$$

Note that $\mathbb{E}[T_i] = \mathbb{E}[Y_i] = 0$. Consequently, we can write out the covariances, for any $i, j$, as follows:

$$\text{Cov}(T_j, Y_i) = \mathbb{E}[T_j Y_i] = (\gamma\lambda + \beta\gamma^2)\mathbb{E}[\Theta_U^2] + \beta\mathbb{E}[\Theta_T^2]$$

$$\text{Cov}(T_j, T_i) = \mathbb{E}[T_j T_i] = \gamma^2\mathbb{E}[\Theta_U^2] + \mathbb{E}[\Theta_T^2]$$

$$\text{Cov}(T_i, Y_i) = \mathbb{E}[T_i Y_i] = (\gamma\lambda + \beta\gamma^2)\mathbb{E}[\Theta_U^2] + (\gamma\lambda + \beta\gamma^2)\mathbb{E}[\varepsilon_U^2] + \beta\mathbb{E}[\Theta_T^2] + \beta\mathbb{E}[\varepsilon_T^2]$$

$$\text{Var}(T_i) = \mathbb{E}[T_i T_i] = \gamma^2\mathbb{E}[\Theta_U^2] + \gamma^2\mathbb{E}[\varepsilon_U^2] + \mathbb{E}[\Theta_T^2] + \mathbb{E}[\varepsilon_T^2]$$

$$\text{Var}(Y_i) = \mathbb{E}[Y_i Y_i] = 2(\lambda^2 + \beta^2\gamma^2)\mathbb{E}[\Theta_U^2] + 2(\lambda^2 + \beta^2\gamma^2)\mathbb{E}[\varepsilon_U^2] + \beta^2\mathbb{E}[\Theta_T^2] + \beta^2\mathbb{E}[\varepsilon_T^2] + \mathbb{E}[\Theta_Y^2] + \mathbb{E}[\varepsilon_Y^2]$$

Now, we look at the numerator in (10) and we want to know when it could be zero since that makes the partial correlation zero:

$$0 = \text{Var}(T_i)\text{Cov}(T_j, Y_i) - \text{Cov}(T_j, T_i)\text{Cov}(T_i, Y_i)$$
$$= \gamma\lambda(\mathbb{E}[\Theta_U^2]\mathbb{E}[\varepsilon_T^2] - \mathbb{E}[\varepsilon_U^2]\mathbb{E}[\Theta_T^2])$$

The solution is given by

$$\sigma_{\Theta_U} = \frac{\sigma_U}{\sigma_T}\sigma_{\Theta_T} \, , \tag{11}$$

where the square root of the second moments are equal to the standard deviations. This is the same equality as demonstrated in the example.

## D.5   Asymptotic Behavior of Partial Correlation

We also look at the partial correlation and ask what happens when the confounder effect sizes $\gamma$ or $\lambda$ become very large. The numerator in (10) grows linearly with respect to both $\gamma$ and $\lambda$, and the other variances can be rewritten as

$$\text{Cov}(T_j, T_i) = \gamma^2\mathbb{E}[\Theta_U^2] + O(1)$$
$$\text{Cov}(T_i, Y_i) = (\gamma\lambda + \beta\gamma^2)(\mathbb{E}[\Theta_U^2] + \mathbb{E}[\varepsilon_U^2]) + O(1)$$
$$\text{Var}(T_i) = \gamma^2(\mathbb{E}[\Theta_U^2] + \mathbb{E}[\varepsilon_U^2]) + O(1)$$
$$\text{Var}(Y_i) = 2(\lambda^2 + \beta^2\gamma^2)(\mathbb{E}[\Theta_U^2] + \mathbb{E}[\varepsilon_U^2]) + O(1)$$

where $O(1)$ is a constant with respect to $\gamma$ and $\lambda$.

We rewrite the partial correlation (8) as

$$\rho_{T_j, Y_i \cdot T_i} = \frac{\left(\text{Var}(T_i)\text{Cov}(T_j, Y_i) - \text{Cov}(T_j, T_i)\text{Cov}(T_i, Y_i)\right)/\sqrt{\text{Var}(T_i)^3\text{Var}(Y_i)}}{\sqrt{1 - \frac{\text{Cov}(T_j, T_i)^2}{\text{Var}(T_i)^2}}\sqrt{1 - \frac{\text{Cov}(T_i, Y_i)^2}{\text{Var}(T_i)\text{Var}(Y_i)}}} \, .$$

Assuming that all second moments are non-zero and finite, it is possible to show that

$$\rho_{T_j, Y_i \cdot T_i} \propto \begin{cases} \gamma^{-3} & \text{for } |\gamma| >> 1, \\ 1 & \text{for } |\lambda| >> 1 \end{cases} \, .$$

Hence, when either $|\gamma|$ or $|\lambda|$ goes to infinity we have

$$\rho_{T_j, Y_i \cdot T_i} \xrightarrow[|\gamma| \to \infty]{} 0$$

$$\rho_{T_j, Y_i \cdot T_i} \xrightarrow[|\lambda| \to \infty]{} C \text{ for some } C \in [-1, 1]$$

Note that $C$ could be zero, for instance, when $\sigma_{\Theta_U} = \frac{\sigma_U}{\sigma_T}\sigma_{\Theta_T}$, although we demonstrate with simulation studies in Appendix F a case where $C$ is non-zero as well.

Interestingly, in this case, the bias from estimating the causal effect without adjusting for the confounder $U$ is

$$\mathbb{E}[Y \mid do(T)] - \mathbb{E}[Y \mid T] = \beta T - (\beta T + \frac{\lambda}{\gamma}T) = -\frac{\lambda}{\gamma}T \, . \tag{12}$$

We note that when $\gamma \to \infty$ the bias goes to zero, similar to the partial correlation. Meanwhile, the bias increases with $\lambda$ which also is consistent with the asymptotic behavior of the partial correlation as $\lambda \to \infty$.

## D.6 Positivity violations in the sampling of mechanisms

In this section, we discuss another potential issue that can come up in our problem setting, namely that there could be positivity violations in the sampling of the mechanisms. Particularly unique to our setting is that the support for $(\Theta_T, \Theta_Y, \Theta_X, \Theta_U)$ must be the same for different environments, if not then this would be a direct violation of Assumption 4.

We start by pointing out that there exist two categories of positivity violations: structural violations and random violations [Hernan and Robins, 2023]. Structural violations occur for instance when a certain range of values of a variable never will be observed, and they may restrict the population for which we can draw causal conclusions. Meanwhile, random violations are due to having a finite number of samples. Random violations are perhaps also less problematic, as they can go away as we collect more data.

In our setting with multi-environment data, we can make the same analogy with structural and random positivity violations. For instance, a structural violation could occur if we have only collected data from multiple hospitals in country A but there is also another country B such that $P(\Theta_T, \Theta_Y, \Theta_X, \Theta_U \mid \text{country A})$ does not overlap with $P(\Theta_T, \Theta_Y, \Theta_X, \Theta_U \mid \text{country B})$. Then, we have a structural violation between environments in countries A and B. Meanwhile, a random positivity violation could come from not observing enough environments in country A, assuming that we are only interested in studying data from that country.

So based on what we have discussed so far, one could ask how to reason about positivity violations when trying to detect hidden confounding. In this case, we think it is important to first ask ourselves: In what population are we trying to detect hidden confounding? If the answer is the population in country A, then we should not include data from country B, as there is a structural positivity violation; and vice versa. For random violations, it is harder to anticipate what problems can come up, but these issues are mainly avoided by ensuring that we have sampled enough environments. This is what, in the end, will affect the quality of the conditional independence test in our method.

## D.7 Selection bias

In our theory, we assume there is no selection bias. That is, for instance, that there are no unobserved colliders that we have conditioned on in our causal DAG. In principle, we can study such scenarios by adding colliders in our original problem setup to introduce different types of selection mechanisms. While we leave this topic for future work, we present here a quick illustration of how selection bias can (or can not) hurt our procedure.

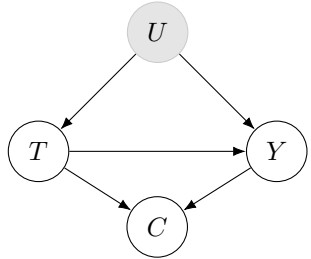

Figure 10: Grey variables are unobserved.

**Example 2.** *Consider the graph $\mathcal{G}$ in Figure 10 where $C$ is a collider between treatment $T$ and outcome $Y$. Now we consider the corresponding augmented DAG $\mathcal{G}^*$, add the causal mechanism $\Theta_C \sim P(\Theta_C)$ as parent to $C$, and check how (unknowingly) conditioning on $C$ influences our ability to detect the presence of the unobserved $U$. The fact that $\Theta_C$ is a random variable would reflect that we have different selection mechanisms in different environments. We note in this case that the conditional independence $T_j^{(k)} \not\perp_P Y_i^{(k)} \mid T_i^{(k)}, C_i^{(k)}, C_j^{(k)}$ will be violated regardless of whether $U$ is present or not. In other words, selection bias would in this case lead to false positives of our procedure. This is because there will always be an open path between $T_j^{(k)}$ and $Y_i^{(k)}$ through $\Theta_C$ in $\mathcal{G}^*$. But if, on the other hand, the selection mechanism remains fixed across all environments (meaning $\Theta_C$ is constant), this would close that path. That means we do not have this problem of false detection anymore, and the proposed approach would still work.*

# E Generation of Twins Semi-synthetic Dataset

We use data from twin births in the USA between 1989-1991 [Almond et al., 2005, Louizos et al., 2017] to construct an observational dataset with a known causal structure. The dataset contains 46 covariates related to pregnancy, birth, and parents, out of which we select ten as potential confounders in our experiments. Many covariates are highly imbalanced and have low variance, some even on the border of being constant. Due to this, we wish to exclude those in data generation. The selection is performed by first excluding any binary variables from the list of covariates, and then further removing the remaining ones that have an empirical variance smaller than one.

In the end, the following covariates are used from the Twins dataset for our data generation: birth month, father's age, mom's age, mom's education, mom's place of birth, number of prenatal visits, number of live births before twins, the total number of births before twins, period of gestation, state of birth occurrence, and state of registered residence. Note that all of these are reported as categorical/discrete variables in the dataset.

Among the covariates, we use the state in which the birth took place as environment label, hence obtaining 51 different environments. We simulate a continuous treatment and outcome by randomly selecting $p \in [1, \ldots, 10]$ features $(X_1, X_2, \ldots, X_p)$ and a set of functions to generate the data as follows:

$$T = \sum_{d=1}^{p} \alpha_d f_d(X_{d,\text{scaled}}) + \varepsilon_T \text{ and } Y = \sum_{d=1}^{p} \beta_d g_d(X_{d,\text{scaled}}) + \delta T + \varepsilon_Y \tag{13}$$

For each $d$, $\alpha_d$ is sampled from a uniform distribution $\text{Unif}(1,5)$, $\beta_d$ is sampled from a uniform distribution $\text{Unif}(1,5)$, and $f_d$ and $g_d$ are sampled from the set of functions $\{\tanh(x), x, x^2\}$ with equal probability. The treatment effect, $\delta$, is also randomly sampled from a uniform distribution $\text{Unif}(1,2)$, and we have noise variables $\varepsilon_T \sim \text{Normal}(0, 1/4)$ and $\varepsilon_Y \sim \text{Normal}(0, 1/4)$. The features from the Twins dataset are also scaled as

$$X_{d,\text{scaled}} = 5 * (X_d - \text{mean}(X_d))/(\text{max}(X_d) - \text{min}(X_d)), \tag{14}$$

where mean/max/min are taken over all observed values of the covariate $X_d$. Note that the functions are fixed across all environments in this setup, and variations between environments only stem from the real-world distribution shifts of the covariates between birth states.

# F Additional Experiments

We present additional simulation studies, mainly replicating the experiments on synthetic data from Section 5.1 with continuous data. In addition, we further investigate the asymptotic behavior of the partial correlation from Appendix D.3 with both the binary and continuous data.

The continuous data is generated from a linear-Gaussian DAG as described in (7). Unless otherwise stated, we use $\beta = 1$, $\sigma_T = \sigma_U = \sigma_Y = 1$, $\sigma_{\Theta_T} = \sigma_{\Theta_Y} = 1$ and $\sigma_{\Theta_T} = 5$. To vary the influence of the hidden confounder, we can adjust either $\gamma$ or $\lambda$. We test $T_j^{(k)} \perp\!\!\!\perp_P Y_i^{(k)} \mid T_i^{(k)}$ using the partial correlation [Baba et al., 2004] with $N_k = 2$ samples per environment.

For the first experiment, we vary the number of environments and the confounder influence by setting $\gamma = \lambda$. In the main part of the paper, we only considered what happens when varying $\lambda$ with $\gamma = 1$. Similarly, we do the same experiment with the binary data for comparison. The results are seen in Figure 11. Notably, the probability of detecting hidden confounding starts decreasing when $\gamma = \lambda$ goes above a certain threshold. This is consistent with our previous conclusions from Appendix D.3, where we noted that partial correlation is proportional to $\gamma^{-3}$ for $\gamma >> 1$ while remaining constant for $\lambda >> 1$. Hence, we would expect the partial correlation to shrink as both $\gamma$ and $\lambda$ grow. Notably, the effect is more pronounced with the continuous data although it can also be seen with the binary data.

Secondly, we perform the experiment with continuous data where we only vary $\lambda$ while fixing $\gamma = 1$. The results are shown in Figure 12, we note that the probability of detection no longer decreases as the

confounder effect size increases. The results with binary data are also shown again for comparison. Once again, this is predicted by the asymptotic behavior of the partial correlation.

Finally, we compare our statistical testing procedure to testing $Y \perp\!\!\!\perp_P E \mid T$ with continuous data in Figure 14. We run the experiment with 10000 environments, 100 samples per environment, and $\sigma_{\Theta_U} = 10$ to avoid the issues of faithfulness violations which we have discussed before. Similar to the case with binary data, in Figure 13b, we see that the probability of false detection when testing $Y \perp\!\!\!\perp_P E \mid T$ grows as the standard deviation of $\Theta_Y$ increases. Meanwhile, we do not observe this problem in our testing procedure. The case when confounding is present is shown in Figure 13a to confirm that our method is able to detect confounding. For completion, we also include a case where $\lambda = 10$ (confounding is present) for the binary data experiment presented in the main part of the paper, this is shown in Figure 14a.

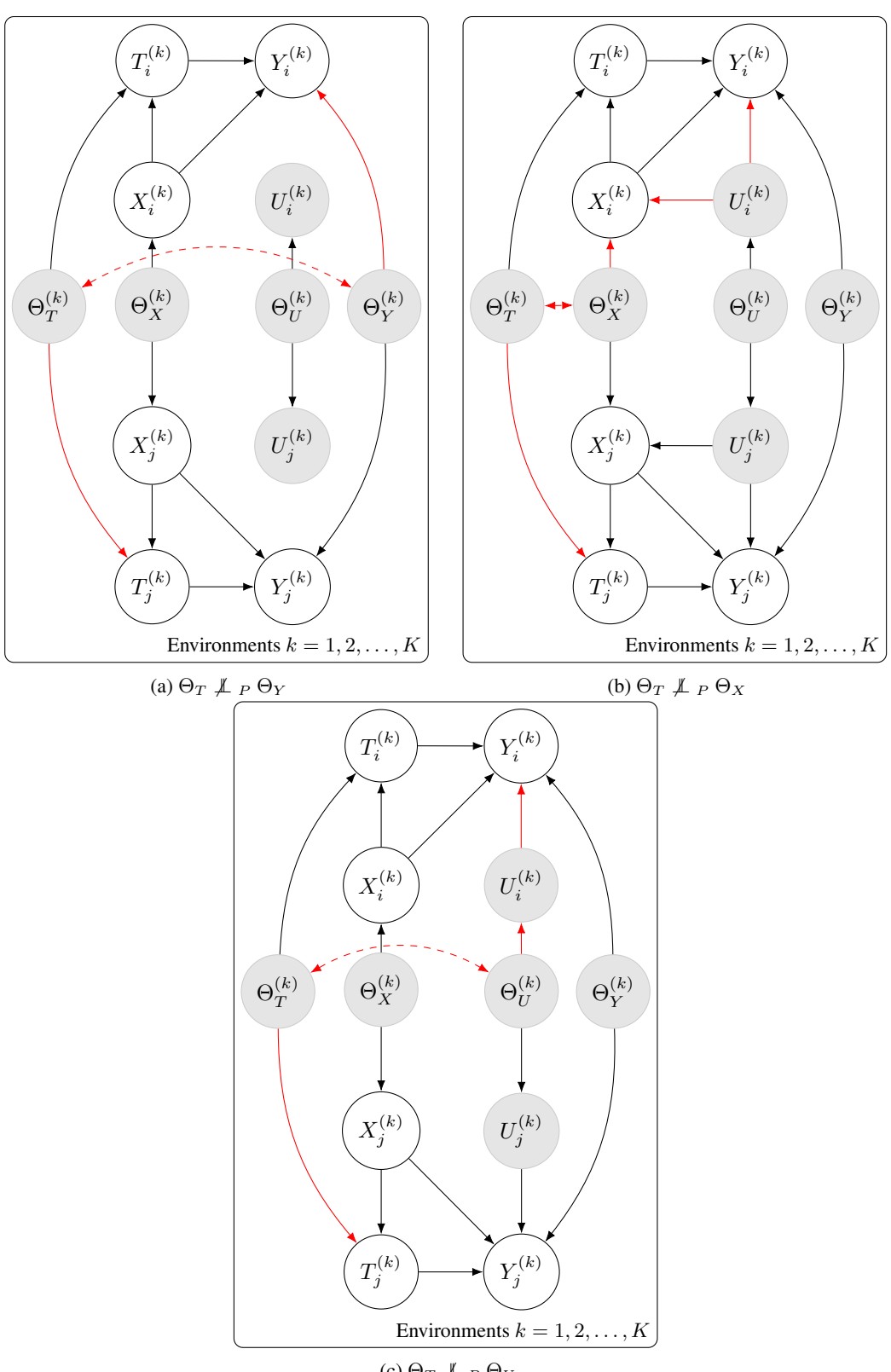

Figure 7: Violations of $T_j^{(k)} \perp\!\!\!\perp_P Y_i^{(k)} \mid T_i^{(k)}, X_i^{(k)}, X_j^{(k)}$ with dependent mechanisms despite that $X$ is a valid adjustment set for estimating $P(Y \mid do(T))$. Open paths between $T_j^{(k)}$ and $Y_i^{(k)}$ after conditioning on $T_i, X_i$ and $X_j$ are marked in red.

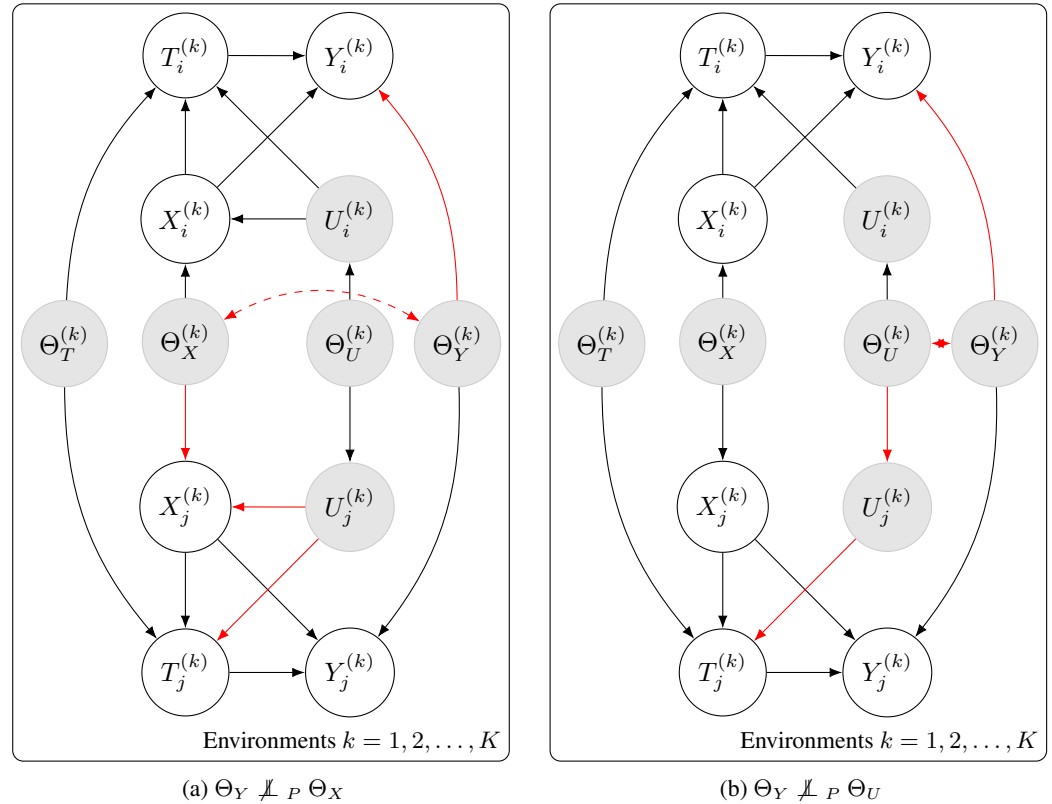

(a) $\Theta_Y \not\perp_P \Theta_X$

(b) $\Theta_Y \not\perp_P \Theta_U$

Figure 8: Violations of $T_j^{(k)} \perp_P Y_i^{(k)} \mid T_i^{(k)}, X_i^{(k)}, X_j^{(k)}$ with dependent mechanisms despite that $X$ is a valid adjustment set for estimating $P(Y \mid do(T))$. Open paths between $T_j^{(k)}$ and $Y_i^{(k)}$ after conditioning on $T_i, X_i$ and $X_j$ are marked in red.

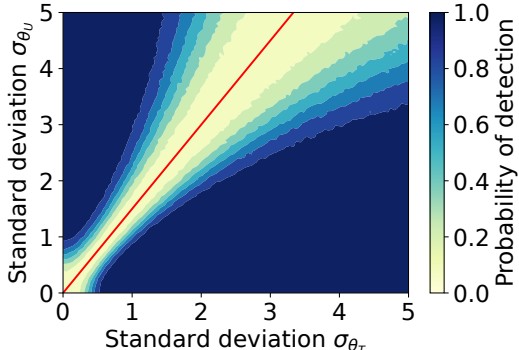

Figure 9: Faithfulness violation from Example 1. Varying $\sigma_{\Theta_T}$ and $\sigma_{\Theta_U}$ with $\sigma_T = \frac{2}{3}$ and $\sigma_U = 1$, the probability for detecting confounding goes to zero as we get closer to $\sigma_{\Theta_U} = \frac{\sigma_U}{\sigma_T} \sigma_{\Theta_T}$ (red line).

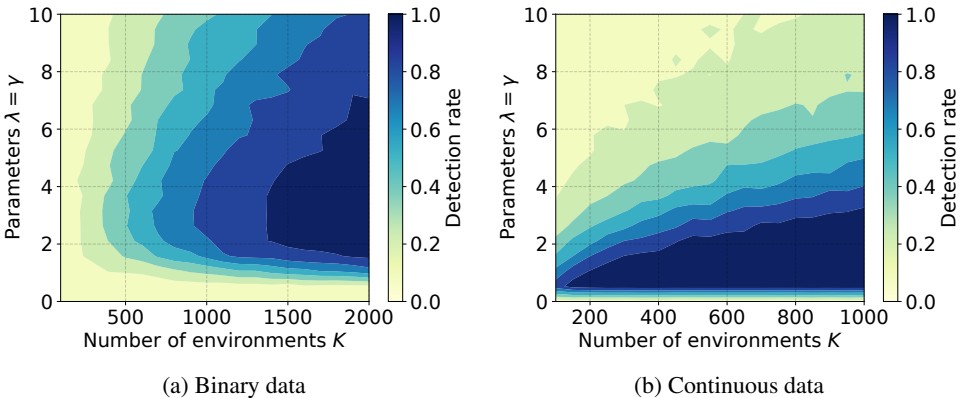

(a) Binary data            (b) Continuous data

Figure 11: Probability of detecting hidden confounding for varying the number of environments $K$ and confounder effect sizes where $\lambda = \gamma$ are varied jointly. 500 repetitions.

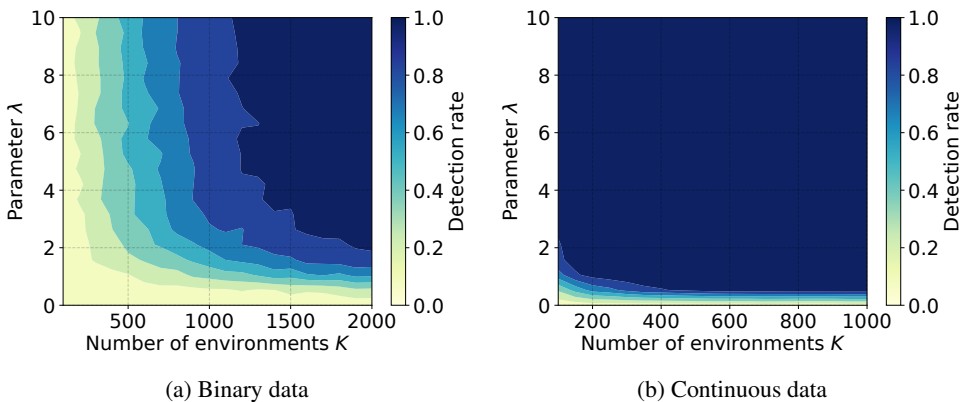

(a) Binary data            (b) Continuous data

Figure 12: Probability of detecting hidden confounding for varying the number of environments $K$ and confounder effect sizes where $\lambda$ is varied while $\gamma = 1$ is fixed. 500 repetitions

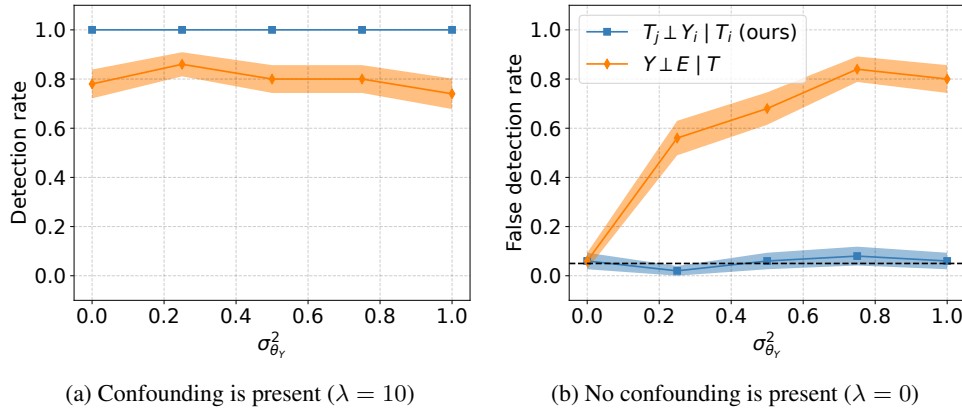

(a) Confounding is present ($\lambda = 10$)       (b) No confounding is present ($\lambda = 0$)

Figure 13: Comparison on continuous linear-Gaussian data between the proposed procedure and an alternative testing procedure by varying the standard deviation of $\Theta_Y$ in both the presence and absence of confounding. The black dashed line corresponds to the desired type 1 error control $\alpha = 0.05$. The shaded area shows the standard error from 50 repetitions.

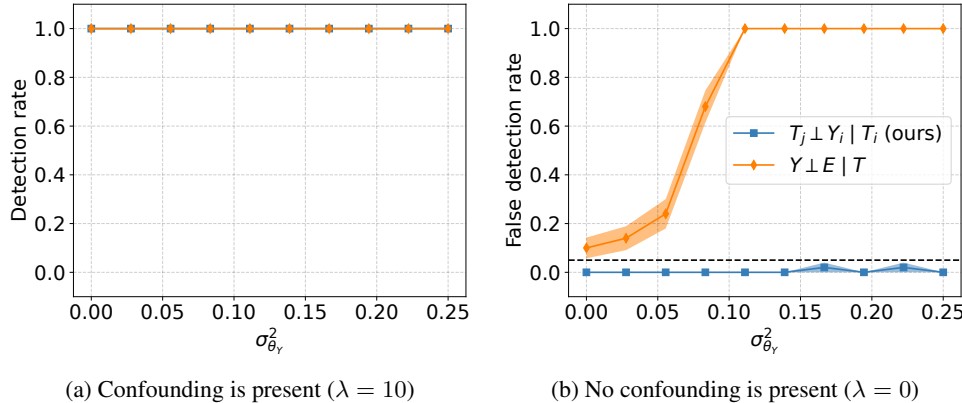

(a) Confounding is present ($\lambda = 10$)  (b) No confounding is present ($\lambda = 0$)

Figure 14: Comparison on binary data between the proposed procedure and an alternative testing procedure by varying the standard deviation of $\Theta_Y$ in both the presence and absence of confounding. The black dashed line corresponds to the desired type 1 error control $\alpha = 0.05$. The shaded area shows the standard error from 50 repetitions.

Table 1: Conditional d-separations in combinations of DAGs with variables $(T, Y, X, U)$. For the d-separation, (✓) indicates that it holds and (✗) otherwise. The shaded rows are the cases where $U$ is a confounder to $T$ and $Y$ in $\mathcal{G}$.

| id | $X-T$ | $X-Y$ | $T-Y$ | $U-T$ | $U-Y$ | $U-X$ | $T_j^{(k)} \perp_d Y_i^{(k)} \mid T_i^{(k)}, X_i^{(k)}, X_j^{(k)}$ | $U$ is confounder |
|---|---|---|---|---|---|---|---|---|
| 1 | → | → | → | → | → | → | ✗ | ✓ |
| 2 | → | → | → | → | → | ← | ✗ | ✓ |
| 3 | → | → | → | → | → |   | ✗ | ✓ |
| 4 | → | → | → | → |   | → | ✓ | ✗ |
| 5 | → | → | → | → |   | ← | ✓ | ✗ |
| 6 | → | → | → | → |   |   | ✓ | ✗ |
| 7 | → | → | → | ← | → | ← | ✓ | ✗ |
| 8 | → | → | → | ← | → |   | ✓ | ✗ |
| 9 | → | → | → | ← | ← | ← | ✓ | ✗ |
| 10 | → | → | → | ← | ← |   | ✓ | ✗ |
| 11 | → | → | → | ← |   | ← | ✓ | ✗ |
| 12 | → | → | → | ← |   |   | ✓ | ✗ |
| 13 | → | → | → |   | → | → | ✓ | ✗ |
| 14 | → | → | → |   | → | ← | ✓ | ✗ |
| 15 | → | → | → |   | → |   | ✓ | ✗ |
| 16 | → | → | → |   | ← | ← | ✓ | ✗ |
| 17 | → | → | → |   | ← |   | ✓ | ✗ |
| 18 | → | → | → |   |   | → | ✓ | ✗ |
| 19 | → | → | → |   |   | ← | ✓ | ✗ |
| 20 | → | → | → |   |   |   | ✓ | ✗ |
| 21 | → | → |   | → | → | → | ✗ | ✓ |
| 22 | → | → |   | → | → | ← | ✗ | ✓ |
| 23 | → | → |   | → | → |   | ✗ | ✓ |
| 24 | → | → |   | → |   | → | ✓ | ✗ |
| 25 | → | → |   | → |   | ← | ✓ | ✗ |
| 26 | → | → |   | → |   |   | ✓ | ✗ |
| 27 | → | → |   | ← | → | ← | ✓ | ✗ |
| 28 | → | → |   | ← | → |   | ✓ | ✗ |
| 29 | → | → |   | ← | ← | ← | ✓ | ✗ |
| 30 | → | → |   | ← | ← |   | ✓ | ✗ |
| 31 | → | → |   | ← |   | ← | ✓ | ✗ |
| 32 | → | → |   | ← |   |   | ✓ | ✗ |
| 33 | → | → |   |   | → | → | ✓ | ✗ |
| 34 | → | → |   |   | → | ← | ✓ | ✗ |
| 35 | → | → |   |   | → |   | ✓ | ✗ |
| 36 | → | → |   |   | ← | ← | ✓ | ✗ |
| 37 | → | → |   |   | ← |   | ✓ | ✗ |
| 38 | → | → |   |   |   | → | ✓ | ✗ |
| 39 | → | → |   |   |   | ← | ✓ | ✗ |
| 40 | → | → |   |   |   |   | ✓ | ✗ |

Table 2: Conditional d-separations in combinations of DAGs with variables $(T, Y, U)$ (this excludes any observed confounder $X$). For the d-separation, (✓) indicates that it holds and (✗) otherwise. The shaded rows are the cases where $U$ is a confounder to $T$ and $Y$ in $\mathcal{G}$.

| id | $T-Y$ | $U-T$ | $U-Y$ | $T_j^{(k)} \perp_d Y_i^{(k)} \mid T_i^{(k)}$ | $T_j^{(k)} \perp_d Y_i^{(k)} \mid Y_j^{(k)}$ | $U$ is confounder | $Y$ ancestor of $T$ |
|---|---|---|---|---|---|---|---|
| 1 | → | → | → | ✗ | ✗ | ✓ | ✗ |
| 2 | → | → |   | ✓ | ✗ | ✗ | ✗ |
| 3 | → | ← | → | ✓ | ✗ | ✗ | ✗ |
| 4 | → | ← | ← | ✓ | ✗ | ✗ | ✗ |
| 5 | → | ← |   | ✓ | ✗ | ✗ | ✗ |
| 6 | → |   | → | ✓ | ✗ | ✗ | ✗ |
| 7 | → |   | ← | ✓ | ✗ | ✗ | ✗ |
| 8 | → |   |   | ✓ | ✗ | ✗ | ✗ |
| 9 | ← | → | → | ✗ | ✗ | ✓ | ✓ |
| 10 | ← | → | ← | ✗ | ✓ | ✗ | ✓ |
| 11 | ← | → |   | ✗ | ✓ | ✗ | ✓ |
| 12 | ← | ← | ← | ✗ | ✓ | ✗ | ✓ |
| 13 | ← | ← |   | ✗ | ✓ | ✗ | ✓ |
| 14 | ← |   | → | ✗ | ✓ | ✗ | ✓ |
| 15 | ← |   | ← | ✗ | ✓ | ✗ | ✓ |
| 16 | ← |   |   | ✗ | ✓ | ✗ | ✓ |
| 17 |   | → | → | ✗ | ✗ | ✓ | ✗ |
| 18 |   | → | ← | ✗ | ✓ | ✗ | ✓ |
| 19 |   | → |   | ✓ | ✓ | ✗ | ✗ |
| 20 |   | ← | → | ✓ | ✗ | ✗ | ✗ |
| 21 |   | ← | ← | ✓ | ✓ | ✗ | ✗ |
| 22 |   | ← |   | ✓ | ✓ | ✗ | ✗ |
| 23 |   |   | → | ✓ | ✓ | ✗ | ✗ |
| 24 |   |   | ← | ✓ | ✓ | ✗ | ✗ |
| 25 |   |   |   | ✓ | ✓ | ✗ | ✗ |

Table 3: Conditional d-separations in combinations of DAGs with variables $(T, Y, X, U)$. We test $T_j^{(k)} \perp\!\!\!\perp_d Y_i^{(k)} \mid T_i^{(k)}, X_i^{(k)}, X_j^{(k)}, Z$ where $Z$ contains the mechanisms that are degenerate, as noted per column. For the d-separation, (✓) indicates that it holds and (✗) otherwise. The shaded rows are the cases where $U$ is a confounder to $T$ and $Y$ in $\mathcal{G}$.

| id | $X-T$ | $X-Y$ | $T-Y$ | $U-T$ | $U-Y$ | $U-X$ | $U$ is confounder | Deg. $\theta_T$ | Deg. $\theta_Y$ | Deg. $\theta_X$ | Deg. $\theta_U$ | Deg. $\theta_T,\theta_Y$ | Deg. $\theta_T,\theta_X$ | Deg. $\theta_T,\theta_U$ | Deg. $\theta_Y,\theta_X$ | Deg. $\theta_Y,\theta_U$ | Deg. $\theta_X,\theta_U$ | Deg. $\theta_T,\theta_Y,\theta_X$ | Deg. $\theta_T,\theta_Y,\theta_U$ | Deg. $\theta_T,\theta_X,\theta_U$ | Deg. $\theta_Y,\theta_X,\theta_U$ | Deg. $\theta_T,\theta_Y,\theta_X,\theta_U$ |
|---|---|---|---|---|---|---|---|---|---|---|---|---|---|---|---|---|---|---|---|---|---|---|
| 1 | → | → | → | → | → | → | ✓ | ✗ | ✗ | ✗ | ✗ | ✗ | ✗ | ✗ | ✗ | ✗ | ✗ | ✗ | ✗ | ✓ | ✗ | ✓ |
| 2 | → | → | → | → | → | ← | ✓ | ✗ | ✗ | ✗ | ✗ | ✗ | ✗ | ✓ | ✗ | ✗ | ✗ | ✗ | ✓ | ✓ | ✗ | ✓ |
| 3 | → | → | → | → | → |  | ✓ | ✗ | ✗ | ✗ | ✗ | ✗ | ✗ | ✓ | ✗ | ✗ | ✗ | ✗ | ✓ | ✓ | ✗ | ✓ |
| 4 | → | → | → | → |  | → | ✗ | ✓ | ✓ | ✓ | ✓ | ✓ | ✓ | ✓ | ✓ | ✓ | ✓ | ✓ | ✓ | ✓ | ✓ | ✓ |
| 5 | → | → | → | → |  | ← | ✗ | ✓ | ✓ | ✓ | ✓ | ✓ | ✓ | ✓ | ✓ | ✓ | ✓ | ✓ | ✓ | ✓ | ✓ | ✓ |
| 6 | → | → | → | → |  |  | ✗ | ✓ | ✓ | ✓ | ✓ | ✓ | ✓ | ✓ | ✓ | ✓ | ✓ | ✓ | ✓ | ✓ | ✓ | ✓ |
| 7 | → | → | → | ← | → | ← | ✗ | ✓ | ✓ | ✓ | ✓ | ✓ | ✓ | ✓ | ✓ | ✓ | ✓ | ✓ | ✓ | ✓ | ✓ | ✓ |
| 8 | → | → | → | ← | → |  | ✗ | ✓ | ✓ | ✓ | ✓ | ✓ | ✓ | ✓ | ✓ | ✓ | ✓ | ✓ | ✓ | ✓ | ✓ | ✓ |
| 9 | → | → | → | ← | ← | ← | ✗ | ✓ | ✓ | ✓ | ✓ | ✓ | ✓ | ✓ | ✓ | ✓ | ✓ | ✓ | ✓ | ✓ | ✓ | ✓ |
| 10 | → | → | → | ← | ← |  | ✗ | ✓ | ✓ | ✓ | ✓ | ✓ | ✓ | ✓ | ✓ | ✓ | ✓ | ✓ | ✓ | ✓ | ✓ | ✓ |
| 11 | → | → | → | ← |  | ← | ✗ | ✓ | ✓ | ✓ | ✓ | ✓ | ✓ | ✓ | ✓ | ✓ | ✓ | ✓ | ✓ | ✓ | ✓ | ✓ |
| 12 | → | → | → | ← |  |  | ✗ | ✓ | ✓ | ✓ | ✓ | ✓ | ✓ | ✓ | ✓ | ✓ | ✓ | ✓ | ✓ | ✓ | ✓ | ✓ |
| 13 | → | → | → |  | → | → | ✗ | ✓ | ✓ | ✓ | ✓ | ✓ | ✓ | ✓ | ✓ | ✓ | ✓ | ✓ | ✓ | ✓ | ✓ | ✓ |
| 14 | → | → | → |  | → | ← | ✗ | ✓ | ✓ | ✓ | ✓ | ✓ | ✓ | ✓ | ✓ | ✓ | ✓ | ✓ | ✓ | ✓ | ✓ | ✓ |
| 15 | → | → | → |  | → |  | ✗ | ✓ | ✓ | ✓ | ✓ | ✓ | ✓ | ✓ | ✓ | ✓ | ✓ | ✓ | ✓ | ✓ | ✓ | ✓ |
| 16 | → | → | → |  | ← | ← | ✗ | ✓ | ✓ | ✓ | ✓ | ✓ | ✓ | ✓ | ✓ | ✓ | ✓ | ✓ | ✓ | ✓ | ✓ | ✓ |
| 17 | → | → | → |  | ← |  | ✗ | ✓ | ✓ | ✓ | ✓ | ✓ | ✓ | ✓ | ✓ | ✓ | ✓ | ✓ | ✓ | ✓ | ✓ | ✓ |
| 18 | → | → | → |  |  | → | ✗ | ✓ | ✓ | ✓ | ✓ | ✓ | ✓ | ✓ | ✓ | ✓ | ✓ | ✓ | ✓ | ✓ | ✓ | ✓ |
| 19 | → | → | → |  |  | ← | ✗ | ✓ | ✓ | ✓ | ✓ | ✓ | ✓ | ✓ | ✓ | ✓ | ✓ | ✓ | ✓ | ✓ | ✓ | ✓ |
| 20 | → | → | → |  |  |  | ✗ | ✓ | ✓ | ✓ | ✓ | ✓ | ✓ | ✓ | ✓ | ✓ | ✓ | ✓ | ✓ | ✓ | ✓ | ✓ |
| 21 | → | → |  | → | → | → | ✓ | ✗ | ✗ | ✗ | ✗ | ✗ | ✗ | ✗ | ✗ | ✗ | ✗ | ✗ | ✗ | ✓ | ✗ | ✓ |
| 22 | → | → |  | → | → | ← | ✓ | ✗ | ✗ | ✗ | ✗ | ✗ | ✗ | ✓ | ✗ | ✗ | ✗ | ✗ | ✓ | ✓ | ✗ | ✓ |
| 23 | → | → |  | → | → |  | ✓ | ✗ | ✗ | ✗ | ✗ | ✗ | ✗ | ✓ | ✗ | ✗ | ✗ | ✗ | ✓ | ✓ | ✗ | ✓ |
| 24 | → | → |  | → |  | → | ✗ | ✓ | ✓ | ✓ | ✓ | ✓ | ✓ | ✓ | ✓ | ✓ | ✓ | ✓ | ✓ | ✓ | ✓ | ✓ |
| 25 | → | → |  | → |  | ← | ✗ | ✓ | ✓ | ✓ | ✓ | ✓ | ✓ | ✓ | ✓ | ✓ | ✓ | ✓ | ✓ | ✓ | ✓ | ✓ |
| 26 | → | → |  | → |  |  | ✗ | ✓ | ✓ | ✓ | ✓ | ✓ | ✓ | ✓ | ✓ | ✓ | ✓ | ✓ | ✓ | ✓ | ✓ | ✓ |
| 27 | → | → |  | ← | → | ← | ✗ | ✓ | ✓ | ✓ | ✓ | ✓ | ✓ | ✓ | ✓ | ✓ | ✓ | ✓ | ✓ | ✓ | ✓ | ✓ |
| 28 | → | → |  | ← | → |  | ✗ | ✓ | ✓ | ✓ | ✓ | ✓ | ✓ | ✓ | ✓ | ✓ | ✓ | ✓ | ✓ | ✓ | ✓ | ✓ |
| 29 | → | → |  | ← | ← | ← | ✗ | ✓ | ✓ | ✓ | ✓ | ✓ | ✓ | ✓ | ✓ | ✓ | ✓ | ✓ | ✓ | ✓ | ✓ | ✓ |
| 30 | → | → |  | ← | ← |  | ✗ | ✓ | ✓ | ✓ | ✓ | ✓ | ✓ | ✓ | ✓ | ✓ | ✓ | ✓ | ✓ | ✓ | ✓ | ✓ |
| 31 | → | → |  | ← |  | ← | ✗ | ✓ | ✓ | ✓ | ✓ | ✓ | ✓ | ✓ | ✓ | ✓ | ✓ | ✓ | ✓ | ✓ | ✓ | ✓ |
| 32 | → | → |  | ← |  |  | ✗ | ✓ | ✓ | ✓ | ✓ | ✓ | ✓ | ✓ | ✓ | ✓ | ✓ | ✓ | ✓ | ✓ | ✓ | ✓ |
| 33 | → | → |  |  | → | → | ✗ | ✓ | ✓ | ✓ | ✓ | ✓ | ✓ | ✓ | ✓ | ✓ | ✓ | ✓ | ✓ | ✓ | ✓ | ✓ |
| 34 | → | → |  |  | → | ← | ✗ | ✓ | ✓ | ✓ | ✓ | ✓ | ✓ | ✓ | ✓ | ✓ | ✓ | ✓ | ✓ | ✓ | ✓ | ✓ |
| 35 | → | → |  |  | → |  | ✗ | ✓ | ✓ | ✓ | ✓ | ✓ | ✓ | ✓ | ✓ | ✓ | ✓ | ✓ | ✓ | ✓ | ✓ | ✓ |
| 36 | → | → |  |  | ← | ← | ✗ | ✓ | ✓ | ✓ | ✓ | ✓ | ✓ | ✓ | ✓ | ✓ | ✓ | ✓ | ✓ | ✓ | ✓ | ✓ |
| 37 | → | → |  |  | ← |  | ✗ | ✓ | ✓ | ✓ | ✓ | ✓ | ✓ | ✓ | ✓ | ✓ | ✓ | ✓ | ✓ | ✓ | ✓ | ✓ |
| 38 | → | → |  |  |  | → | ✗ | ✓ | ✓ | ✓ | ✓ | ✓ | ✓ | ✓ | ✓ | ✓ | ✓ | ✓ | ✓ | ✓ | ✓ | ✓ |
| 39 | → | → |  |  |  | ← | ✗ | ✓ | ✓ | ✓ | ✓ | ✓ | ✓ | ✓ | ✓ | ✓ | ✓ | ✓ | ✓ | ✓ | ✓ | ✓ |
| 40 | → | → |  |  |  |  | ✗ | ✓ | ✓ | ✓ | ✓ | ✓ | ✓ | ✓ | ✓ | ✓ | ✓ | ✓ | ✓ | ✓ | ✓ | ✓ |

