# OpenReview forum: "Detecting hidden confounding in observational data using multiple environments"
_NeurIPS.cc/2023/Conference — NeurIPS 2023 poster_

### Official Review · Reviewer_g1qg · 2023-07-06

**Soundness:** 3 good
**Presentation:** 3 good
**Contribution:** 2 fair
**Rating:** 5
**Confidence:** 3

**Summary:**

The authors develop a test for hidden confounding using only observational data from multiple environments with the same DAG under the independent causal mechanisms assumption.

**Strengths:**

The paper is well-written and covers the related work well.
Their setting, assumptions, and contributions are clear and they meaningfully discuss the profound limitations of their work with potential solutions.
Under new (but strong) assumptions, they propose a test for no unmeasured confounding and provide an algorithm using only observational data. The proofs in the construction of the test and the test itself is novel and can be counted as original contributions.
They support their claims through adequate (semi)-synthetic experiments.

**Weaknesses:**

1. My main concerns about the paper is about its practical utility.
    - It is not often a practitioner has access to multiple observational datasets
    - The observational datasets may be highly heterogeneous, e.g., the type of data recorded, the way they are recorded, etc., making the preprocessing task for those datasets to be used in a way that is proposed in this manuscript very hard, if not impossible.
    - Analyzing observational studies is hard for a lot of reasons. As we increase the number of studies, the chances of introducing bias through other and often overlooked ways increase such as non-adherence to treatment assignments, or selection bias introduced while defining the treatment assignments and where the follow-up begins etc. Those can all be reasons behind rejection beyond confounding. Please correct me if I am wrong in that regard.

2. I find it hard to believe that different observational datasets will have the same DAG.
    - Different environments (e.g. hospitals) may have very different mechanisms (e.g. institutional practices) that result in different DAGs.

3. In summary, although the paper formalizes a framework where we can test the no unmeasured confounding assumption, I think the approach would only work in very controlled settings.


**Questions:**

I do not have any question beyond some points in the weaknesses part.

**Limitations:**

The authors acknowledge some important limitations page 6.

---

> ### Author Rebuttal · Authors · 2023-08-04
>
> We thank the reviewer for their insightful questions, as well as pointing out the novelty of designing the test in our paper. We further agree that it is important to keep in mind that our test relies on new strong assumptions that should not be taken for granted.
>
> ## The real-world practical utility of our approach (*see global rebuttal reply*)
>
> As a very similar question also was raised by reviewer 7Ey5 on the real-world motivation of our problem setting, we have written a joint reply found in the global author rebuttal reply. We kindly ask reviewer g1qg to read it there.
>
> ## Having the same DAG in multiple observational datasets
>
> An important assumption is that each environment shares the same DAG. We gave an example of this with the multi-level study from Leite et al. (2015) where a shared structure is assumed across clusters. In this case, such an assumption does not come across as particularly controversial. Looking more broadly, having an invariant structure is a common assumption in causal inference literature, see for instance Barenboim and Pearl (2016) and Rothenhäusler et al. (2021). In these works, the DAG stays the same across different datasets/environments, and the influence of changing environment is modeled by adding it as a variable in the shared DAG. We do agree with the reviewer that it is good to think about this assumption carefully, but we also remain positive that this assumption can oftentimes be made, since the structure of the graph often encodes invariant properties, such as which variable precedes which other variable in time, while the exact mechanism itself is likely to change across environments. It is these variations in the mechanisms that we exploit to construct our method.
>
> ## References
> - Bareinboim, Elias, and Judea Pearl. "Causal inference and the data-fusion problem." Proceedings of the National Academy of Sciences 113.27 (2016): 7345-7352.
> - Leite, W. L., Jimenez, F., Kaya, Y., Stapleton, L. M., MacInnes, J. W., & Sandbach, R. (2015). An evaluation of weighting methods based on propensity scores to reduce selection bias in multilevel observational studies. Multivariate behavioral research, 50(3), 265-284.
> - Rothenhäusler, Dominik, et al. "Anchor regression: Heterogeneous data meet causality." Journal of the Royal Statistical Society Series B: Statistical Methodology 83.2 (2021): 215-246.

---

> > ### Comment · Reviewer_g1qg · 2023-08-19
> > **Response to Rebuttal**
> >
> > I thank the authors for their response. I have read the other reviews and responses as well, and decided to keep my original score.

---

### Official Review · Reviewer_7Ey5 · 2023-07-07

**Soundness:** 3 good
**Presentation:** 4 excellent
**Contribution:** 3 good
**Rating:** 6
**Confidence:** 3

**Summary:**

The authors focus on a setting where we observe treatments, outcomes, and a set of confounders across multiple environments.  While it's generally not possible to identify unobserved confounders in a single data set, when we have multiple data sets and assume the mechanisms operate in the same way across those environments (even if the parameters may differ), we can test for the presence of unobserved confounders.  The authors propose an independence test between a treatment in one environment and an outcome in another, and show that, if they are not independent (conditioned on the observed confounders and the other treatment), then there must be at least one unobserved confounder that's producing that dependence.

**Strengths:**

The presentation of this paper is excellent, and I found the explanation throughout very clear.  The idea comes across as fairly simple, which speaks to the clarity of the explanation.   The graphics in particular (Figures 1 and 2) go a long way towards making the approach understandable

I appreciate the attention paid to assumptions, including the discussion in 4.1 about assumption relaxation.

**Weaknesses:**

The main weakness I see of this paper is the lack of motivation for the setting the authors consider.  This work is focused on a situation where we have observational data from multiple environments where we have collected the same variables from all environments and, while parameters are allowed to differ between them, the actual mechanisms at play need to have the same structure.  The authors bring up the example in the intro of different hospitals administering the same treatment but serving different populations, but otherwise, I feel like this work would be served by a discussion of how realistic this sort of setting is, or how common it is in the real world.

Similarly, while I do like Section 4.1 and the authors' attention to the assumptions, the lack of clear real-world ties extends here.  What would it look like in real data if we had pair-wise dependent mechanisms?  Is this something we would expect to see frequently?  This lack of discussion makes it hard to assess how important these assumptions actually are, and how useful the relaxations are in turn.

Considering that Section 4.2 is the actual core of the approach, I wish more attention were paid to it.  I understand that space is limited, but including the actual algorithm you're proposing seems worth moving something else to the appendix.

**Questions:**

While being able to detect whether or not there is hidden confounding is useful, it would be more useful to then make some statement about the strength of hidden confounding, or how much we might expect it to affect our estimates.  I suspect in many realistic situations, there is going to be /some/ hidden confounding.  If care is taken when choosing which potential confounders to measure, though, the effect of this hidden confounding is hopefully small. Do you think your approach could be used or modified to answer questions about the strength of hidden confounding?

**Limitations:**

The authors' approach is confined to a fairly specific setting, where the same variables, operating in the same manner (with different parameters), are measured across different environments.  The approach also only outputs a binary yes or no for whether hidden confounding exists.  If the test comes back positive for hidden confounding, it's unclear if this means causal analysis is impossible, or what the next steps should be.

In terms of societal impact, I don't see any real potential for negative impact.  This work can be used to test for one potential hurdle for accurate causal inference (hidden confounders), but there are many other assumptions that need to be met as well, and this paper does a good job at making those clear.

---

> ### Author Rebuttal · Authors · 2023-08-04
>
> We thank reviewer 7Ey5 for their valuable feedback on our work. As suggested by them, in the final version of the paper, we intend to bring the algorithm back to the main paper if space allows in the camera-ready version. Similarly, we will also extend the introduction with two additional examples of use cases for our method.
>
> ## The real-world practical utility of our approach (*see global rebuttal reply*)
>
> As a very similar question also was raised by reviewer g1qg on the real-world motivation of our problem setting, we have written a joint reply found in the global author rebuttal reply. We kindly ask reviewer 7Ey5 to read it there.
>
> ## Quantifying the strength of hidden confounding
>
> We agree that it would be a great addition to go beyond a yes/no answer on the presence of hidden confounding and instead quantity the magnitude of the bias incurred by hidden confounding. We mention this possibility in our future works section in the appendix. While doing so is a much more difficult task, one empirical observation makes us believe our method can be modified to quantity the strength of the hidden confounding. Figure 4a in the paper shows that the test statistic from the conditional independence test increases if the bias from hidden confounding goes up. Thus, perhaps this test statistic can act as a proxy for the size of the bias from hidden confounding. For instance, a simple heuristic procedure to gauge the effect of hidden confounding would involve eliminating one or more observed confounders from the analysis and seeing the resulting variations in both the test statistic and the estimated causal effect. This would give us a calibrated guess on how the magnitude of the statistic relates to the bias. By then evaluating the test statistic when all observed confounders are accounted for, we might get a clue what the corresponding remaining bias could be. Turning such ideas into a principled method requires significantly more research, which we are currently engaged in.

---

### Official Review · Reviewer_ZLeH · 2023-07-07

**Soundness:** 2 fair
**Presentation:** 2 fair
**Contribution:** 2 fair
**Rating:** 4
**Confidence:** 4

**Summary:**

The authors proposed a detection method for hidden confounders, with only observational data in multiple environments. They, in particular, designed a testable independence condition for it, under the assumptions of (i) Faithfulness & Causal Markov Property, (ii)Shared Causal Graph, (iii) Independent Causal Mechanism Principle, and (iv) Non-degenerate Probabilistic Independent Causal Mechanisms. They also performed synthetic and semi-synthetic experiments to verify the effectiveness of their proposed method.

**Strengths:**

This paper is well written with clear motivation and a description of their method.

What the authors focused on is indeed an interesting yet challenging research topic in causal inference. And the idea of using conditional independence conditions, especially between observations, is interesting to me.

**Weaknesses:**

The problem setting has some confusing notations.

The experiments cannot verify better performances than other methods for detecting latent confounders.


**Questions:**

Regarding the problem setting, there are some confusing notations. For example, since the $E^{(k)}$consists of four causal mechanisms shown in Fig.1(a), how could it serve as an indicator variable for what environment observations belong to? Btw, why $E^{(k)}$ is observed while the $(\theta_T^{(k)} , \theta_Y^{(k)}, \theta_X^{(k)}, \theta_U^{(k)} )$ are all unobserved, whereas $E = (\theta_T^{(k)} , \theta_Y^{(k)}, \theta_X^{(k)}, \theta_U^{(k)} )$.

In line 116, the authors said, “We do not assume to know anything about the individual parameters $(\theta_T, \theta_Y, \theta_X, \theta_U )$, however in their method, they sampled the mechanisms i.i.d. for $(\theta_T, \theta_Y, \theta_X, \theta_U )$.

In line 113, ${T, Y, X, Y}$ might be ${T, Y, X, U}$.

In Figure 1, $\theta_T, \theta_Y, \theta_X, \theta_U$ are parents of the ${T, Y, X, U}$.respectively. So I suggest removing the subscript $\theta_V$ in Eq.2 since it’s easily misleading for the readers that $\theta_V$ is the parameter of conditional distributions.

Regarding the theory, Theorems 1 and 2 seem to derive the independence condition from a fixed environment $k$ other than using the advantages from multiple environments, as shown in Eq.(3). In contrast, the authors used multiple environments for the independent testing.

Since the goal is to detect latent confounders, I wonder if some other baselines for a single environment could obtain better results. We might see multiple environments as one environment for this verification.

In line 289, the distribution notation for the $\theta_V^{(k)}$ should have reflected the heterogeneity.


**Limitations:**

The needed number of environments is so high.

---

> ### Author Rebuttal · Authors · 2023-08-04
>
> We thank reviewer ZLeH for their helpful feedback. We have corrected the typo on line 113 pointed out by them. For clarity, we will address their questions in a Q&A format below.
>
> **Q: Since the $E^{(k)}$ consists of four causal mechanisms shown in Fig.1(a), how could it serve as an indicator variable for what environment observations belong to? Btw, why is $E^{(k)}$ observed while the $(\Theta_T^{(k)}, \Theta_Y^{(k)}, \Theta_X^{(k)}, \Theta_U^{(k)})$ are all unobserved, whereas $E^{(k)}=(\Theta_T^{(k)}, \Theta_Y^{(k)}, \Theta_X^{(k)}, \Theta_U^{(k)})$.**
>
> A: In each environment, denoted with $E^{(k)}$, we have the parameters $(\Theta_T^{(k)}, \Theta_Y^{(k)}, \Theta_X^{(k)}, \Theta_U^{(k)})$ for the conditional distributions (i.e. causal mechanisms) in eq. (2) in the paper. Importantly, we are saying that for a specific environment, the distribution $P(T,Y,X,U \mid \Theta_T^{(k)}, \Theta_Y^{(k)}, \Theta_X^{(k)}, \Theta_U^{(k)})$ is fixed as the parameters are fixed. These parameters themselves are not observed, in the sense that their values are unknown. For simplicity we wrote $E^{(k)}=(\Theta_T^{(k)}, \Theta_Y^{(k)}, \Theta_X^{(k)}, \Theta_U^{(k)})$ to underline their connection. But we understand how using an equality sign here might be confusing. For this reason, we have removed this equality statement in the text and Figure 1a, and included a better explanation of their relationship in the original paragraph (lines 112 - 118).
>
> **Q: In line 116, the authors said, “We do not assume to know anything about the individual parameters $(\Theta_T^{(k)}, \Theta_Y^{(k)}, \Theta_X^{(k)}, \Theta_U^{(k)}$”. However in their method, they sampled the mechanisms i.i.d. for $(\Theta_T^{(k)}, \Theta_Y^{(k)}, \Theta_X^{(k)}, \Theta_U^{(k)})$.**
>
> A: We will clarify what we mean by “not knowing anything” in this sentence. In particular, our approach does not require that we observe the values of these parameter values $(\Theta_T^{(k)}, \Theta_Y^{(k)}, \Theta_X^{(k)}, \Theta_U^{(k)})$. We do not even try to infer them in our method. The only assumption we make about them is that they are pairwise independent random variables with non-degenerate distributions. We will edit line 116 to make this clearer.
>
> **Q: In Figure 1,  $(\Theta_T^{(k)}, \Theta_Y^{(k)}, \Theta_X^{(k)}, \Theta_U^{(k)}$ are parents of the $(T,Y,X,U)$ respectively. So I suggest removing the subscript $\Theta_V$ in Eq. (2) since it’s easily misleading for the readers that $\Theta_V$ is the parameter of conditional distributions.**
>
> A: We believe this is a misunderstanding, as this is exactly how we intend them to be interpreted.
>
> **Q: Regarding the theory, Theorems 1 and 2 seem to derive the independence condition from a fixed environment $k$ other than using the advantages from multiple environments, as shown in Eq.(3). In contrast, the authors used multiple environments for the independent testing.**
>
> A: Let’s clarify what the independence condition in Theorem 1 says (the same explanation also applies to Theorem 2). Within a given environment $E^{(k)}$ if there is hidden confounding, Theorem 1 states: the outcome of individual $i$ should be dependent on the treatment given to another individual $j$ even after adjusting for all other available information. This condition, while relating two individuals in the same environment, holds for all $k$. To better understand why multiple environments are necessary, we can look at the following example. If we only had a single environment, then the data would always be i.i.d. as the parameters are fixed within a single environment. Thus, the condition in Theorem 1 would fail because individuals always would be independent of each other – regardless of whether there is hidden confounding or not.
>
> **Q: Since the goal is to detect latent confounders, I wonder if some other baselines for a single environment could obtain better results. We might see multiple environments as one environment for this verification.**
>
> A: It is well known that without additional assumptions in a single environment, it is impossible to detect hidden confounding (see, for instance, Peters et al, 2017). Our key research question is whether access to data from multiple heterogeneous environments allows us to detect hidden confounding. Keeping this question in mind, the strongest baseline we found was the Joint Causal Inference framework by Mooij et al (2020) which, similar to ours, also uses conditional independence testing with multi-environment data and makes no further parametric assumptions. Still, we do mention other methods in the related work section, such as Janzing and Schölkopf (2018) that can be applied in a single environment setting. Their approach, however, assumes linear relations with a one-dimensional real-valued confounder. These are strong assumptions that deviate significantly from our setting, making it difficult to construct a fair comparison. With that said, we are happy to hear if the reviewer has specific suggestions for other suitable baselines.
>
> **Q: In line 289, the distribution notation for the $\Theta_V^{(k)}$ should have reflected the heterogeneity.**
>
> A: We believe this is already reflected as $\Theta_V^{(k)}$ is Normal-distributed across environments. We are happy to further discuss how to clarify this for the reviewer if they can elaborate on their question.
>
> ## References
> - Janzing, Dominik, and Bernhard Schölkopf. "Detecting confounding in multivariate linear models via spectral analysis." Journal of Causal Inference 6.1 (2018): 20170013.
> - Mooij, Joris M., Sara Magliacane, and Tom Claassen. "Joint causal inference from multiple contexts." The Journal of Machine Learning Research 21.1 (2020): 3919-4026.
> - Peters, Jonas, Dominik Janzing, and Bernhard Schölkopf. Elements of causal inference: foundations and learning algorithms. The MIT Press, 2017.

---

> > ### Comment · Reviewer_ZLeH · 2023-08-16
> > **Additional questions**
> >
> > Thanks for your responses. I still have the following concerns.
> >
> > 1. Regarding "Since this condition, while relating two individuals in the same environment, holds for all k, then in a single environment, such two individuals still have this dependence relation."  Why have i.i.d. relations？
> >
> > 2. Regarding baselines. Why aren't there options like FCI or similar methods with fewer assumptions considered here?  Additionally, there are notable papers that could potentially enrich the discussion and facilitate comparisons. For instance, the following works might warrant attention:
> >
> > Ghassami, A. E., Kiyavash, N., Huang, B., et al. (2018). 'Multi-domain causal structure learning in linear systems.' Advances in Neural Information Processing Systems, 31.
> >
> > Huang, B., Zhang, K., Gong, M., et al. (2020). 'Causal discovery from multiple data sets with non-identical variable sets.' Proceedings of the AAAI Conference on Artificial Intelligence, 34(06), 10153-10161."
> >
> >
> > If there are any misconceptions about the points I've raised above, then I welcome input from the authors.

---

> > > ### Author Response · Authors · 2023-08-17
> > > **Reply to additional questions**
> > >
> > > Regarding your first question: we interpret it as you asking why we have i.i.d. data within a given environment. The answer is that this follows from our assumptions as, for a given environment $k$,  $P(T,Y,X,U \mid \Theta_T^{(k)}, \Theta_Y^{(k)}, \Theta_X^{(k)}, \Theta_U^{(k)})$ is a fixed distribution that we sample independently from. Thus, for a given $k$, the data is i.i.d.. Notice however that we can have dependency relationships between observations in the same environment $k$ when not conditioning on the parameters $\Theta_T^{(k)}, \Theta_Y^{(k)}, \Theta_X^{(k)}, \Theta_U^{(k)}$; this is what we see in Theorem 1 and 2.
> > >
> > > For the second concern, we thank you for the additional suggestions for baselines. Let's start by addressing why we haven't compared our approach to a standard causal discovery algorithm like FCI. Our primary objective is not to learn a complete causal DAG. Instead, we aim to detect hidden confounding while having some prior knowledge of the underlying structure. Additionally, although FCI is a natural consideration, it does not work in our context. With the ground-truth graph in Figure 2 from our paper, there are no observed conditional independencies. Thus, FCI outputs an uninformative PAG that gives no information on whether we can exclude the presence of a hidden confounder between $T$ and $Y$. As we discuss in section C in the Appendix of our paper, there are graphs similar to the one in Figure 2 where ordinary constraint-based algorithms might be able to detect hidden confounding, but these do not cover all possible cases of our problem setting.
> > >
> > > Further, regarding the papers you mentioned: both are relevant to our work and deserve inclusion in our related works section, which we will address. For instance, Ghassami et al. (2018) also leverage the independent causal mechanism principle for structure learning. However, neither of these papers are considering the same problem as us. Ghassami et al. (2018) assume, in contrast to us, that all variables are observed (i.e. causal sufficiency) which means that their method can not be used for detecting hidden confounding. Meanwhile, Huang et al. (2020) assume constant linear relationships across environments: $b_{ij}$ appears fixed in (1) in their paper. In addition to assuming linearity unlike us, this constant relationship assumption represents a stringent constraint not imposed within our framework.

---

### Official Review · Reviewer_1FZJ · 2023-07-08

**Soundness:** 3 good
**Presentation:** 3 good
**Contribution:** 3 good
**Rating:** 7
**Confidence:** 4

**Summary:**

This paper presents a method for detecting hidden confounding in observational data.  Without further assumptions, detecting hidden confounding is known to be impossible.  This paper takes advantage of a common scenario where one has access to observational data collected from multiple environments.  Given particular assumptions that some of the underlying mechanisms and/or exogenous factors are varying across environments, this paper presents a method to detect hidden confounding between a pair of target variables <T,Y>.  The paper presents the theory, a discussion of its assumptions and which can be weakened, synthetic experiments and one experiment in a real-world dataset.


**Strengths:**

- detecting hidden confounding under reasonable assumptions is a useful contribution.

- the paper is well written and easy to understand.  I appreciate the additional detail on assumptions in sec 4.1.

- experiments demonstrate implications in synthetic settings and 1 real-world dataset

**Weaknesses:**

- I don't think this approach will handle detection of hidden confounding under selection bias (i.e., where a data collection mechanism means not all data samples are visible in an environment).  This is a common challenge in practice, and it is worth mentioning this as a limitation and/or noting it in the initial problem statement.

- the paper addresses a scenario where causal mechanisms are parameterized by environment variables $\theta_{T,X,U,V,..}$.  If the causal mechanisms themselves are varying, I'm not sure I understand what the motivating causal inference scenario is.  That is, if the causal inference question is to identify $P(Y|T)$, how is that a well-defined question if the causal mechanism defining $Y|Pa(Y)$ is changing across environments?

- the paper would be improved with additional real world datasets.

**Questions:**


- The current causal graph representation of the problem seems like it cannot represent situations where selection bias is occurring.  I.e., where we only observe X,Y,T when certain conditions hold and essentially throw away the data if those conditions do not hold.  Is my understanding correct?

- Assuming that the parameters $\theta_{T,Y,X,U}$ vary across environments is a strong assumption.  I appreciate the discussion of a violation of this assumption starting in line 237.  I'd recommend adding a forward-reference to this discussion weakening assumptions.

- There are various positivity assumptions here (e.g., in sampling causal mechanisms and in sampling data within an environment).  In practice, such positivity assumptions are violated often.  E.g., we may gather many sets of data in country A where some variable may take only a small range of values.  Later, we may realize there is hidden confounding when we gather new data sets in country B and the given variable takes on a larger range values and, for example, passes some threshold.  I am concerned that a naive user of your method will over rely on your method's analysis of data from country A and believe that there is no hidden confounding that will threaten their inferred causal inference mechanism even in country B.  Note that this is distinct from the discussion section 4.1 in lines 237-245 --- even if P(T|Pa(T)) is varying --- even if an RCT is performed in country A --- it would not identify that effects will be different in our hypothetical country B.   It would be good to highlight this limitation, e.g., as another paragraph in 4.1


**Limitations:**


The authors have largely addressed limitations and broader impacts.  I appreciate the note of caution regarding use of this work in high-stakes settings.

---

> ### Author Rebuttal · Authors · 2023-08-04
>
> We thank reviewer 1FZJ for their insightful feedback and questions. Based on their recommendation, to help future readers of our paper, we have added an earlier reference to our discussion on the violation of our assumptions when first presenting the assumptions in the paper.
>
> ## The influence of selection bias in our theory
>
> 1FZj raises an important point that we have not commented on what would happen under selection bias. Indeed, in our current setup, we assume there is no selection bias, and have not discussed this further in the paper. We will clarify this in our problem description to avoid confusion for future readers. Still, it is interesting to reason how our procedure would be influenced by selection bias. To do so, we can add colliders in our original graph in the paper to introduce different types of selection mechanisms. As a quick illustration of how selection bias can (or can not) hurt our procedure, we provide an example: say there is a selection mechanism that acts as a collider between treatment and outcome. Further, let this selection mechanism be changing across environments – maybe we have different conditions for when data is thrown out in each environment. Then, we expect our procedure to falsely detect hidden confounding even when it is not present. But if, on the other hand, the selection mechanism remains fixed across all environments, we do not have this problem of false detection anymore, and the proposed approach would still work. We will add this example to the appendix, where we also discuss the influence of other assumption violations.
>
> ## What is the causal inference question if $P(Y\mid Pa(Y))$ is changing across environments?
>
> This is an attentive question, as it is always good to carefully think about the causal estimand of interest. First of all, as discussed in section 4.1, it should be noted that it is not necessary for $P(Y\mid Pa(Y))$ to change across environments for our procedure to work. Meaning, it works whether $P(Y\mid Pa(Y))$ changes or not. Therefore, one could very well assume that $P(Y\mid Pa(Y))$ (or at least $E[Y\mid Pa(Y)]$) is fixed – as in a standard causal inference setting.
>
> We can however also give an example where  $P(Y\mid Pa(Y))$ changes with the environment $E$ and we still have well-defined estimands. Say that our environments are the different states of the US, like in the Twins dataset we use in our experiments. One possible causal estimand here would be state-specific interventions, that is $P(Y\mid do(T=t), E)$. This goes more in the direction of attempting to capture treatment effect variation, which is natural to do in a hierarchical model (Feller and Gelman, 2015). We could however also country-wide interventions $P(Y\mid do(T=t))$ – meaning we aggregate the state-specific effects into one country-wide effect. For both of these estimands, our method still helps us determine if the estimand is identifiable. Because in both cases we require that there is no unmeasured confounding within each environment.
>
> ## Violation of positivity assumption
>
> We thank reviewer 1FZJ for bringing up this discussion on positivity violations. Related to this discussion, we think it also highlights the need to define a population of interest. In practice, positivity violations are common and can be divided into two categories: structural violations and random violations (Hernán and Robins, 2020). Structural violations occur for instance when a certain range of values of a variable never will be observed, and they may restrict the population for which we can draw causal conclusions. Meanwhile, random violations are due to having a finite number of samples. Random violations are perhaps also less problematic, as they can go away as we collect more data.
>
> In our setting with multi-environment data, we can make the same analogy with structural and random positivity violations. If we understand the setting you are describing correctly, you are describing a structural violation where some types of environments are never observed in country A. Thus, before using our method, we think it is important to ask ourselves: in what population are we trying to detect if there is hidden confounding? If the answer is the population in country A, then we should not use data from country A to also make conclusions about country B, and vice versa. For random violations, it is hard to anticipate what problems can come up, but we can avoid them by having enough environments from the population of interest.
>
> While positivity violations are not a unique limitation to our method, we agree that it is important to highlight how to reason about them in our problem setting. This would encourage responsible usage of our method. We are adding a paragraph on this in the appendix, in the same section as where we discuss the other assumption violations.
>
> ## References
> - Feller, Avi, and Andrew Gelman. "Hierarchical models for causal effects." Emerging Trends in the Social and Behavioral Sciences: An interdisciplinary, searchable, and linkable resource (2015): 1-16.
> - Hernán Miguel A., Robins James M. Causal Inference: What If. Boca
> Raton: Chapman & Hall/CRC. (2020)

---

> > ### Comment · Reviewer_1FZJ · 2023-08-15
> >
> > Thank you for your clarifications and careful consideration and incorporation of feedback.

---

### Author Rebuttal · Authors · 2023-08-04

This is a joint reply to reviewers 7Ey5 and g1qg for their questions on the real-world practical utility of our proposed method. For the camera-ready version, if space allows, we will include the below examples in the introduction of our paper to help future readers understand our method’s practical utility.

## The real-world practical utility of our approach

We believe reviewers 7Ey5 and g1qg both bring up important points of addressing how realistic or common our proposed setting is. Furthermore, as mentioned by g1qg, analyzing multiple observational datasets can be very hard. Especially when it comes to pre-processing and taking into account other sources of bias. Yet, there are many settings where such analyses are done and below we give some examples; these also describe scenarios where we believe our method would be suitable to use.

Firstly, note that environments don't need to come from completely different studies: they could also be pre-existing clusters within a study. Particularly, we believe our method is applicable in many existing multi-level studies in which individuals are nested in clusters and nonrandomly assigned to a treatment/control on an individual level. In this type of setting it is natural to introduce a hierarchical model, like the one we consider in our paper. One example is found in Leite et al. (2015): they are interested in estimating the effect of giving special education services to first-graders on their reading performance in the third grade. They analyze data with 7,783 students (i.e. individuals) clustered across 1454 schools (i.e. environments). In this case, they use a hierarchical model that shares the same structure for all schools, and they include random parameters to allow changing causal mechanisms between schools.

Another example setting is individual participant data meta-analyses. This is a type of analyses that uses all individual-level data from multiple studies instead of aggregating summary statistics (Riley et al. 2010). They are resource intensive but also do enable us to draw stronger conclusions. While individual participant data meta-analyses can vary in size, Di Angelantonio (2016) offers one example where 239 prospective studies are jointly analysed. This showcases a setting where many observational datasets are analyzed jointly and where our method could be used.

We hope these two examples draw a better picture of the real-world use cases of our proposed method. For the camera-ready version, we will refer to both of these examples in the introduction of our paper to help future readers understand our method’s practical utility.

## References
- Di Angelantonio, Emanuele, et al. "Body-mass index and all-cause mortality: individual-participant-data meta-analysis of 239 prospective studies in four continents." The Lancet 388.10046 (2016): 776-786.
- Leite, W. L., Jimenez, F., Kaya, Y., Stapleton, L. M., MacInnes, J. W., & Sandbach, R. (2015). An evaluation of weighting methods based on propensity scores to reduce selection bias in multilevel observational studies. Multivariate behavioral research, 50(3), 265-284.
- Riley, Richard D., Paul C. Lambert, and Ghada Abo-Zaid. "Meta-analysis of individual participant data: rationale, conduct, and reporting." Bmj 340 (2010).

---

### Decision · Program_Chairs · 2023-09-21

**Decision:**

Accept (poster)

**Comment:**

The paper proposes a new approach for detecting hidden confounding when multiple datasets are available, sharing some of their causal structure. This is done without performing full causal discovery, a daunting and often impossible task. The paper gives some compelling use cases, and good empirical evidence for the performance of the method. This is a novel and useful research approach, and I expect substantial future work will build on the results presented here.